# $A^2$-FLOW: ALIGNMENT-AWARE PRE-TRAINING FOR SPEECH SYNTHESIS WITH FLOW MATCHING

## ABSTRACT

Recent advances in speech synthesis have enabled highly natural and speaker-adaptive speech generation by leveraging large-scale transcribed datasets. However, requiring tens of thousands of hours of annotated speech is impractical in low-resource settings. Existing pre-trained speech models often utilize masked speech inpainting for pre-training and show strong performance on various speech generation tasks using limited task-specific data. Nonetheless, these models still require external alignment mechanisms or extensive additional training to learn alignment for alignment-aware tasks, such as text-to-speech (TTS). In this paper, we propose $A^2$-Flow, an alignment-aware pre-training method for flow matching models in speech synthesis. $A^2$-Flow integrates alignment learning directly into the pre-training process using discrete speech units, enabling the model to efficiently adapt to alignment-aware tasks without the need for separate alignment mechanisms. By embedding alignment learning into pre-training, $A^2$-Flow facilitates alignment-free voice conversion (VC) and allows for faster convergence during TTS fine-tuning, even with limited transcribed data, making it highly suitable for low-resource scenarios. Experimental results show that $A^2$-Flow superior zero-shot VC performance compared to existing models and matches state-of-the-art TTS performance using only a small amount of transcribed data. Moreover, we demonstrate that $A^2$-Flow can be more efficiently applied to alignment-aware speech synthesis tasks than existing pre-training methods, providing a practical and scalable solution for high-quality speech synthesis across diverse settings.

## 1 INTRODUCTION

Large-scale speech synthesis models have shown exceptional performance in generating highly natural and expressive speech across a wide range of emotions and voice styles, achieving impressive zero-shot text-to-speech (TTS) capabilities even with minimal reference audio (Wang et al., 2023; Kim et al., 2024; Le et al., 2023; Kharitonov et al., 2023; Eskimez et al., 2024). Many non-autoregressive zero-shot TTS models (Shen et al., 2024; Kim et al., 2023b; Le et al., 2023) follow the approach of early non-autoregressive models (Ren et al., 2019; Kim et al., 2020) by using a phoneme duration predictor to align phonemes with speech. This approach simplifies the training of generative decoders by allowing them to focus solely on speech generation rather than alignment modeling. However, the naturalness of the generated speech becomes highly dependent on the performance of the duration predictor, and this approach limits the generative decoder's ability to capture richer information beyond alignment.

Recent non-autoregressive TTS approaches have aimed to eliminate the need for external alignment models by learning alignment jointly with speech generation (Lee et al., 2024; Lovelace et al., 2024; Gao et al., 2023; Eskimez et al., 2024). Among these, E2TTS (Eskimez et al., 2024) uses a flow matching decoder to jointly model text-speech alignment and speech generation, resulting in highly natural prosody and improved generalization to new speakers. However, similar to other zero-shot TTS models, E2TTS relies on a large amount of transcribed data, which is often unavailable in low-resource settings. SpeechFlow (Liu et al., 2024) mitigates the need for large transcribed datasets by using untranscribed speech data during pre-training and generalizes well across tasks such as TTS with minimal fine-tuning. However, it does not learn alignment during pre-training, making it less effective for alignment-aware TTS models like E2TTS, as shown in our experiments.

In this work, we introduce $A^2$-Flow, an alignment-aware pre-training method that integrates discrete HuBERT (Hsu et al., 2021) units into E2TTS's training framework to learn alignment between unit sequences and speech frames. By using de-duplicated units that retain only phonetic content, $A^2$-Flow effectively learns alignment without relying on external duration models. This allows for direct application to zero-shot voice conversion, where phonetic content can be transferred to the target speaker's voice without additional fine-tuning. For TTS tasks, we fine-tune $A^2$-Flow with transcribed data to jointly learn text-speech alignment, eliminating the need for separate alignment modules. To further enhance pronunciation accuracy, we introduce a simple timestep shifting strategy, improving text alignment early in the sampling process and enhancing overall pronunciation accuracy.

Our experimental results demonstrate that our pre-training model successfully captures the alignment between unit sequences and speech frames. By leveraging this pre-training approach, $A^2$-Flow efficiently learns text-speech alignment with 500 hours of data and a few fine-tuning iterations, achieving high pronunciation accuracy. Even with a limited amount of transcribed data, $A^2$-Flow outperforms many existing zero-shot TTS models and achieves comparable results to E2TTS. Moreover, we show that $A^2$-Flow can effectively learn text-speech alignment also for other languages, demonstrating its capability as a multilingual pre-training method. This highlights the flexibility and scalability of $A^2$-Flow, making it a robust foundation for multilingual TTS systems in low-resource settings. Additionally, without any fine-tuning, the pre-trained model itself outperforms existing zero-shot voice conversion models by a large margin, further validating the effectiveness of our pre-training approach.

## 2 METHOD

In this section, we explain the masked speech modeling approach using flow matching as described in Section 2.1. We discuss how this framework is utilized by 3 different models (Voicebox, Speech-Flow, and E2TTS), and outline the motivation for our proposed method. In Section 2.2, we present the pre-training method of $A^2$-Flow and describe its application to voice conversion and TTS.

### 2.1 BACKGROUND: FLOW MATCHING-BASED MASKED SPEECH MODELING

The proposed framework leverages flow matching to model the distribution of mel-spectrograms for various speech synthesis tasks. The core idea is to transform a sample $x_0$ from a simple prior distribution $p_0(x)$ into a data distribution $q(x)$ through a time-dependent vector field $v_t$.

#### 2.1.1 PROBLEM FORMULATION: FLOW MATCHING

Given a mel-spectrogram $x \in \mathbb{R}^{D \times T}$, we define a flow $\phi_t(x)$ parameterized by $v_t$, which describes how $x_0$ evolves into $x$ over time $t \in [0, 1]$ as follows:

$$\frac{d\phi_t(x)}{dt} = v_t(\phi_t(x); \theta, c), \quad \phi_0(x) = x_0, \tag{1}$$

where $v_t$ is the vector field estimated by the model, $\theta$ represents the model parameters, and $c$ is an optional conditioning input that varies based on the task. The model is optimized by minimizing the optimal transport conditional flow matching (OT-CFM) loss:

$$L_{\text{CFM}}(\theta) = \mathbb{E}_{t \sim U[0,1], q(x_1), p_0(x_0)} \left\| u_t(\phi_{t,x_1}(x_0)|x_1) - v_t(\phi_{t,x_1}(x_0); \theta, c) \right\|^2, \tag{2}$$

where $\phi_{t,x_1}(x)$ is the optimal transport conditional flow path, and $u_t(x|x_1)$ is the conditional vector field for each data sample $x_1 \sim q(x)$. The optimal transport conditional flow path can be defined as:

$$\phi_{t,x_1}(x) \sim \mathcal{N}(tx_1, (1 - (1 - \sigma_{\min})t)^2 I), \tag{3}$$

where $\sigma_{\min}$ is a small constant. The target conditional vector field $u_t(x|x_1)$ can then be written as:

$$u_t(x|x_1) = \frac{x_1 - (1 - \sigma_{\min})x}{1 - (1 - \sigma_{\min})t}. \tag{4}$$

By incorporating the conditional probability path and target vector field into the OT-CFM loss in Eq. 2, we can reformulate the objective as:

$$L_{\text{CFM}}(\theta) = \mathbb{E}_{t,x_1,x_0} \left\| v_t(\phi_{t,x_1}(x_0); \theta, c) - u_t(\phi_{t,x_1}(x_0)|x_1) \right\|^2, \tag{5}$$

which encourages the estimated vector field $v_t$ to match the ground truth vector field $u_t$.

The OT-CFM objective allows the model to estimate the marginal vector field $u_t(x)$, which interpolates between the prior distribution $p_0(x)$ and the data distribution $q(x)$. As a result, the model learns to generate data samples by solving the ordinary differential equation in Eq. 1 using the learned vector field.

### 2.1.2 MASKED SPEECH MODELING FRAMEWORK

To apply flow matching to masked speech modeling, we first randomly mask regions of the mel-spectrogram $x$, using a binary mask $m \in \{0, 1\}^{D \times T}$, and define the masked input as $x_{\text{mask}} = (1 - m) \odot x$. Our masking strategy follows Voicebox (Le et al., 2023), where between 70% to 100% of the mel-spectrogram is randomly masked, with a 10% probability of fully masking the entire input.

During training, the masked mel-spectrogram $x_{\text{mask}}$ is concatenated with a noisy mel-spectrogram $\phi_{t,x_1}(x_0)$ along the channel dimension for each timestep $t$. The model is then trained to inpaint the masked regions using the surrounding context based on the following modified CFM loss:

$$L_{\text{masked-CFM}}(\theta) = \mathbb{E}_{t \sim U[0,1], q(x_1), p_0(x_0)} \left\| m \odot \left( u_t(\phi_{t,x_1}(x_0)|x_1) - v_t(\phi_{t,x_1}(x_0); \theta, c) \right) \right\|^2, \quad (6)$$

where $c$ represents all conditioning inputs, including $x_{\text{mask}}$.

During inference, given a reference speech $x^{\text{ref}}$, we concatenate $x^{\text{ref}}$ with a masked region along the temporal axis to form $x_{\text{mask}}^{\text{ref}}$, which serves as the conditioning input. The model then fills in the masked region of the mel-spectrogram using the estimated vector field, taking into account the speaker information from the reference speech. This enables the model to perform zero-shot speaker adaptation based on the reference speech.

### 2.1.3 CONDITIONING INPUT FOR DIFFERENT MODELS

The conditioning input $c$ plays a crucial role in guiding the generative model during training and inference. Below, we outline the different conditioning inputs used in Voicebox (Le et al., 2023), SpeechFlow (Liu et al., 2024), and E2TTS (Eskimez et al., 2024).

**Voicebox** Voicebox is a zero-shot TTS model that uses the masked mel-spectrogram $x_{\text{mask}}$ along with a sequence of aligned phoneme transcripts as the conditioning input $c$. During training, ground truth alignments are used, while during inference, estimated phoneme durations from a separate phoneme duration predictor are used to perform alignment and enable zero-shot TTS.

**SpeechFlow** SpeechFlow is a pre-training method that uses only the masked mel-spectrogram $x_{\text{mask}}$ as the conditioning input $c$ during pre-training and does not employ any additional conditioning. During fine-tuning, task-specific conditioning inputs can be added, such as using aligned phoneme transcripts for zero-shot TTS, similar to Voicebox.

**E2TTS** E2TTS is a zero-shot TTS model that uses $x_{\text{mask}}$ and an unaligned text input $y$ as conditioning inputs $c$. To match the length of $y$ to the speech input, filler tokens are padded at the end of $y$. E2TTS learns the alignment between the unaligned text input and speech without requiring a separate duration modeling module, distinguishing itself from Voicebox's approach.

By using unaligned text input as a conditioning input, E2TTS jointly models text-speech alignment without explicitly learning phoneme alignment, allowing it to generate more natural speech with a simpler architecture compared to Voicebox. However, E2TTS still requires a large amount of paired text-speech data for effective training. Although E2TTS uses 200,000 hours of untranscribed data for pre-training (Wang et al., 2024), it still requires tens of thousands of hours of paired text-speech data and additional training iterations to learn alignment.

Similarly, while SpeechFlow is pre-trained using only masked speech modeling without any conditioning, it can serve as a good initialization point for tasks that do not require explicit alignment learning, such as Voicebox. However, it remains unclear whether SpeechFlow can effectively perform joint alignment and masked speech modeling with limited data under the E2TTS framework.

In Fig. 2, we experimentally demonstrate that SpeechFlow struggles to learn alignment when trained with limited transcribed data under the E2TTS training framework.

These limitations of existing methods highlight the need for a pre-training approach that not only leverages untranscribed data effectively but also facilitates alignment-aware learning. In the next section, we introduce $A^2$-Flow, an alignment-aware pre-training method that specifically addresses these challenges.

## 2.2 $A^2$-FLOW

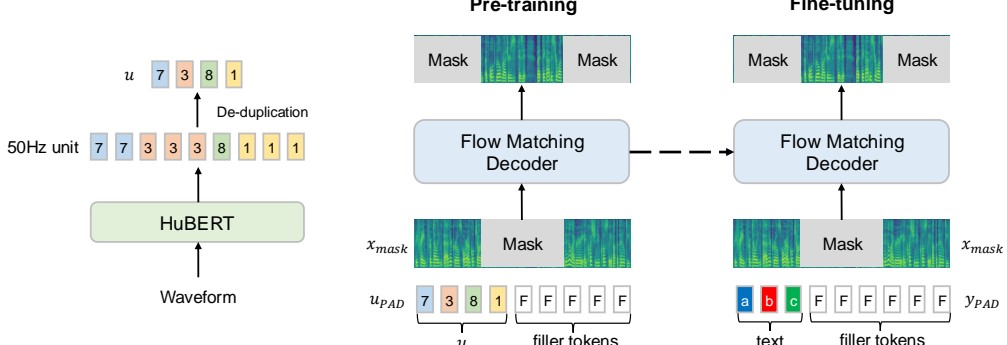

Figure 1: An overview of $A^2$-Flow. During pre-training, $A^2$-Flow utilizes discrete speech unit sequences extracted from HuBERT, allowing the model to learn unit-speech alignment. In the fine-tuning stage, the unit sequences are replaced with text sequences from transcribed data.

We introduce $A^2$-Flow, an alignment-aware pre-training method that utilizes discrete HuBERT units (Hsu et al., 2021). This approach integrates masked speech modeling and alignment learning, thereby making untranscribed speech data useful for downstream tasks like TTS. By jointly learning masked speech and alignment, $A^2$-Flow facilitates the alignment process during fine-tuning and enables more efficient text-speech alignment learning, even with a small amount of transcribed data. This results in TTS systems that can synthesize highly natural and intelligible speech without the need for external alignment mechanisms.

To achieve alignment learning during the pre-training stage, we incorporate discrete HuBERT units—self-supervised speech representations obtained from untranscribed data—into the training process. These units primarily capture phonetic content, enabling the model to learn alignment between the units and the corresponding speech frames. For joint learning of alignment and masked speech modeling, we directly adopt the E2TTS training approach, where text inputs are replaced with discrete unit representations during pre-training, as depicted in Fig. 1.

**Alignment-Aware Pre-Training** Our strategy for alignment-aware pre-training involves removing alignment information from discrete speech units, allowing the model to relearn unit alignment directly during pre-training. To begin, we extract continuous speech representations from 16kHz speech waveforms using the HuBERT model and generate a 50Hz discrete unit sequence $u_{50\text{Hz}}$ using HuBERT's k-means quantizer. This 50Hz unit sequence represents speech as a sequence of indices ranging from 0 to $K-1$, where $K$ is the number of clusters in the k-means quantizer. As shown in Fig. 1, these 50Hz unit sequences often contain repeated indices for continuous speech segments. Since $u_{50\text{Hz}}$ is already aligned with the speech, we remove these repetitions to eliminate duration information, resulting in a de-duplicated unit sequence $u = [u_1, ..., u_L]$.

In $A^2$-Flow, we adopt a similar strategy to E2TTS by padding the de-duplicated unit sequence $u$ with filler tokens $F$ to match the length of the corresponding mel-spectrogram $x$, creating a padded sequence $u_{\text{PAD}} = [u_1, ..., u_L, F, ..., F]$. This padded sequence is concatenated with $x_{\text{mask}}$ and used as part of the conditioning input $c$ to predict the masked regions by optimizing the training objective defined in Eq. 6. During pre-training, $A^2$-Flow uses the de-duplicated units as additional context to guide the inpainting of the masked regions. This approach helps the model learn to extract speaker-specific and acoustic characteristics from the surrounding regions while aligning the de-duplicated units with the corresponding speech.

For multilingual pre-training, we include language IDs as an additional input to the decoder to distinguish between languages within the shared discrete unit space. This allows $A^2$-Flow to better capture language-specific characteristics and more effectively model each language independently.

**Voice Conversion**  Unlike pre-training methods (Liu et al., 2024; Wang et al., 2024), $A^2$-Flow can be directly used for voice conversion tasks without requiring any additional fine-tuning. By leveraging discrete HuBERT units that capture detailed phonetic content, $A^2$-Flow can effectively convert the source speaker's speech into the target speaker's voice, achieving high-quality outputs with minimal adjustment.

For voice conversion, we extract de-duplicated unit sequences $u^{src}$ and $u^{ref}$ from the source and target speech, respectively, and concatenate the mel-spectrogram of the target reference speech $x^{ref}$ with a mask of the source length. Next, we concatenate the extracted de-duplicated sequences $u = [u_{ref}; u_{src}]$ and pad them with filler tokens to match the combined length of $x^{ref}$ and the masked source region. Feeding this sequence into the pre-trained flow matching model, the model generates speech for the masked region by combining the content of $u_{src}$ (source speech) with the speaker characteristics of $x^{ref}$ (target reference speech). This results in voice-converted speech that aligns the source content with the target speaker's voice.

Through alignment-aware pre-training, $A^2$-Flow can generate speech samples from the de-duplicated unit sequence, where duration information has been removed, rather than relying on the original 50Hz unit representation. This flexibility allows $A^2$-Flow to perform voice conversion with varied alignments, providing more diverse and natural outputs compared to conventional voice conversion methods that depend solely on the original unit sequence.

**Fine-tuning For Text-to-Speech**  To perform TTS using the pre-trained $A^2$-Flow, we initialize text embeddings for TTS and fine-tune the pre-trained model using transcribed data with the same objective as in E2TTS. As shown in Fig. 1, we condition the flow matching decoder on the padded text sequence $y_{PAD}$, where the text sequence $y$ is padded with the filler tokens to match the length of the speech. We also train a transformer-based total length predictor to estimate the overall speech duration based on the reference mel-spectrogram and input text. The predictor takes the text sequence $y$ and a short random chunk of the mel-spectrogram $x_{cut}$ as inputs and is trained to estimate the log-scale of the speech length $d$ divided by a scaling factor $s$ using an L2 regression loss.

For zero-shot TTS, given a target reference audio $x^{ref}$ and the transcript $y$ to generate, we first use the total length predictor to estimate the total length of speech corresponding to $y$. We then concatenate $x^{ref}$ with a mask of the predicted length to obtain $x_{mask}^{ref} = [x^{ref}; Mask]$. After concatenating the target reference transcript $y^{ref}$ with the transcript $y$, we pad it to match the length of $x_{mask}^{ref}$. The flow matching decoder then generates samples for the masked regions by solving Eq. 1.

**Sampling**  During the sampling process, we apply classifier-free guidance (CFG) (Ho & Salimans, 2021) by adjusting the vector field estimated through flow matching according to the CFG scale $\gamma$. Additionally, we modify the sampling process to employ a timestep shifting technique inspired by Esser et al. (2024) to sample more values of $t$ closer to 0. This approach ensures that the noisy mel-spectrogram $x_t$ for smaller values of $t$ remains better aligned with the text, resulting in higher pronunciation accuracy.

If the ODE is traditionally solved using uniformly spaced timesteps $t_n = \frac{n}{N}, n = 0, ..., N - 1$, we modify it by non-linearly shifting $t_n$ as defined in Eq. 7, where $\hat{t}_n$ is computed as follows:

$$\hat{t}_n = t_n/(1 + (\alpha - 1) * (1 - t_n)), \tag{7}$$

for a given $\alpha \geq 1$. When $\alpha = 1$ this corresponds to uniform sampling, while larger values of $\alpha$ result in more frequent sampling of $t$ near 0. In our experiments, we set $\alpha = 3$ as the default value. The process of sampling using timestep shifting is explained in more detail in Section A.2.2

## 3  EXPERIMENTS

In this section, we describe the experimental setup, including model architecture, data, training details, and the baselines used for evaluation. We also provide a detailed explanation of the evaluation metrics and methods for both voice conversion and text-to-speech (TTS) tasks.

**Model Architecture**   We employ a modified Diffusion Transformer (DiT) (Peebles & Xie, 2023) architecture by removing the 2D patchify layers. Our model configuration is identical to the DiT-L variant, using a decoder with a hidden size of 1024, 16 attention heads, and a total of 24 transformer layers, resulting in 450M parameters. The model integrates Adaptive LayerNorm (AdaLN) to incorporate embeddings of both the flow matching timestep $t$ and the language ID used during multi-lingual pre-training, allowing for effective conditioning on these factors.

During the pre-training phase, we extract discrete unit sequences using the HuBERT-Large model (Hsu et al., 2021), trained on 220K hours of multilingual data. The unit sequences are obtained using a k-means quantizer trained on the Expresso dataset (Nguyen et al., 2023) with $K = 2000$ clusters. We utilize pre-trained checkpoints made available through the textlesslib library[1].

**Data**   For pre-training, we use a total of 40K hours of speech data, which includes 37K hours of English data from LibriTTS (Zen et al., 2019), LibriVox (Kearns, 2014) and Multilingual LibriSpeech (MLS) (Pratap et al., 2020) datasets, combined with 3K hours of multi-lingual data from the CML dataset, covering seven different languages. To fine-tune the pre-trained model for TTS, we use the transcribed data from the LibriTTS dataset for English, and transcribed data from three other languages (German, French, and Spanish) in the CML dataset to build separate TTS models for each language. The CML transcribed data includes 1500 hours of German, 440 hours of Spanish, and 280 hours of French speech. All speech data is resampled to 22kHz. We convert each 22kHz waveform into an 80-bin log-scale mel-spectrogram using a window length of 1024, hop length of 256, and frequency range of $f_{\min} = 0$ to $f_{\max} = 11025$. These mel-spectrograms serve as the data $x$ used for flow matching in our experiments.

**Training and Fine-tuning**   During pre-training, we use AdamW optimizer with a learning rate of $1e^{-4}$. We train the model for a total of 700K iterations on 32 A100 GPUs, with a batch size of 4 per GPU. For fine-tuning on the TTS task, we initialize the model with the pre-trained weights, and the optimizer is re-initialized. We fine-tune the pre-trained model separately on the transcribed data from LibriTTS and CML-German, CML-French, and CML-Spanish datasets. For all fine-tuning processes, we lower the learning rate to a peak value of $2e^{-5}$, which is reached over 5000 iterations using a warm-up schedule. We fine-tune the pre-trained model for 150K iterations on 8 A100 GPUs, with a batch size of 4 per GPU. After reaching the peak learning rate, it is linearly decayed to zero over the remaining iterations.

**Inference**   During inference, we solve the ordinary differential equation using the Euler method, with 32 sampling steps and a default classifier-free guidance scale of $\gamma = 2$. For voice conversion, we set the default timestep shifting parameter to $\alpha = 1$ to perform uniform timestep sampling, while for TTS, we use $\alpha = 3$ to improve pronunciation accuracy. Once the mel-spectrograms are generated by the flow matching decoder, we convert them into 22kHz waveforms using a BigVGAN-based vocoder (Lee et al.)

**Baselines for Voice Conversion**   We compare the performance of A²-Flow against three voice conversion baselines: Any-to-Any VC (Kovela et al., 2023), UnitSpeech (Kim et al., 2023a), and SelfVC (Neekhara et al., 2024) on the LibriSpeech test-clean dataset. The Any-to-Any VC model is trained on speakers from the test-clean dataset, making it a suitable many-to-many voice conversion baseline that has already learned from reference speakers. UnitSpeech performs speaker adaptation by fine-tuning for 500 iterations on a given reference audio, which makes it a fine-tuning-based voice conversion baseline. Lastly, SelfVC, a recently proposed zero-shot voice conversion model, has been shown to outperform several voice conversion models in zero-shot settings, making it a strong benchmark for comparison.

**Baselines for Text-to-Speech**   We use several models as baselines for zero-shot TTS, including Voicebox, CLaM-TTS, DiTTO-TTS, E2TTS, and SpeechFlow fine-tuned specifically for the TTS task. The results of each model are reported using the evaluation metrics provided in their respective papers for direct comparison in zero-shot TTS scenarios. Additionally, we re-implement E2TTS and SpeechFlow, training both using the same amount of data and identical model architecture as A²-Flow with 32 GPUs over 700K iterations. Note that, unlike the pre-training of SpeechFlow and

---

[1] https://github.com/facebookresearch/textlesslib

$A^2$-Flow, E2TTS is trained with transcripts for the entire 40K hours of data. To evaluate the performance of E2TTS under limited data conditions, we train E2TTS on 500 hours of LibriTTS data using 8 GPUs. We refer to this model as E2TTS-LT. For the re-implemented versions of E2TTS, E2TTS-LT, and SpeechFlow, we apply our timestep shifting scale to $\alpha = 3$, ensuring that comparisons exclude improvements due to differences in the sampling method. This setup allows us to systematically compare the alignment learning and performance of $A^2$-Flow, SpeechFlow, and E2TTS under various training configurations.

**Evaluation and Metrics**    We evaluate zero-shot TTS performance using two evaluation tasks following the approach described in (Wang et al., 2023; Le et al., 2023). The first task is the continuation task, where for all test samples between 4 to 10 seconds long, we provide the first 3 seconds of audio along with the entire transcript and generate speech beyond the initial 3 seconds. The second task is the cross-reference synthesis task, where for each test sample between 4 to 10 seconds long, we use the transcript of the sample and a randomly selected 3-second segment from another sample of the same speaker as the reference audio for zero-shot TTS. For each task, the generated samples are evaluated by measuring speaker similarity to the 3-second reference audio using a speaker similarity metric, and pronunciation accuracy is measured using the word error rate (WER) between the ground truth transcript and the transcript obtained from an ASR model applied to the generated audio.

To evaluate voice conversion performance, we use all samples between 4 to 10 seconds long in the test set. Each sample is used as the source audio, and a 3-second segment is randomly extracted from a different speaker's sample to serve as the reference audio. The model then generates speech that matches the content of the source audio while adopting the voice characteristics of the reference speaker. We evaluate the pronunciation accuracy and speaker similarity of the generated speech compared to the reference audio. Performance is reported as the average WER and SECS values across 1130 pairs from the LibriSpeech test-clean set.

For the speaker similarity metric, we follow (Wang et al., 2023) and use a WavLM-based speaker verification model (Chen et al., 2022) to map both samples to speaker embeddings and measure the cosine similarity between them. We measure speaker similarity between the reference ground truth audio and the generated audio, as defined as SECS-O in the Voicebox (Le et al., 2023). For pronunciation accuracy, we also follow (Wang et al., 2023) and use the same HuBERT-L-based ASR model (Hsu et al., 2021) to measure the WER of English-generated speech, and for other languages, we use the Whisper-large v2 (Radford et al., 2022) model to measure WER.

For subjective evaluation, we generate zero-shot TTS samples using $A^2$-Flow and compare them against samples downloaded from the demo pages of SpeechFlow and E2TTS. We conduct A/B tests with 100 human evaluators, asking them to choose the sample that more closely matches the reference audio in terms of prosody, emotion, and timbre. Each evaluator performs 9 A/B tests when comparing with SpeechFlow and 19 A/B tests when comparing with E2TTS.

## 4    RESULTS

### 4.1    ALIGNMENT-AWARE PRE-TRAINING DYNAMICS

In this section, we analyze the training dynamics of $A^2$-Flow during pre-training and its impact on text-speech alignment during TTS fine-tuning. We compare $A^2$-Flow with E2TTS and SpeechFlow, highlighting differences in alignment performance across training iterations.

For pre-training evaluation, we measure the model's ability to reconstruct masked regions using the unit sequence of the full utterance. We mask speech segments except for the first 3 seconds of each sample in the LibriTTS test-clean set and measure the pronunciation accuracy (WER) and speaker similarity (SECS-O) of generated samples. Fig. 3 in Appendix demonstrates that $A^2$-Flow achieves a WER of 3% after 100K iterations, effectively learning text-speech alignment during pre-training. As training progresses, the model maintains its alignment ability and further enhances speaker similarity.

To show the effectiveness of our alignment-aware pre-training approach in the context of TTS tasks that require joint modeling of text and speech alignment, we compare $A^2$-Flow with SpeechFlow

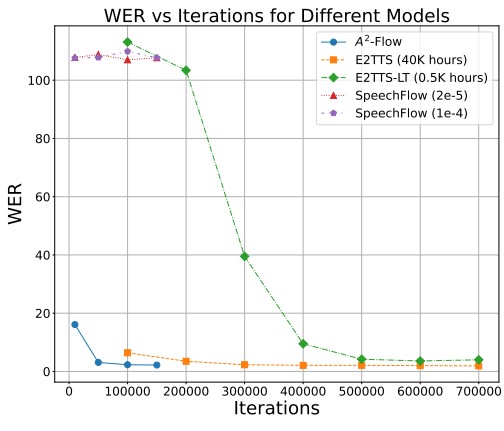
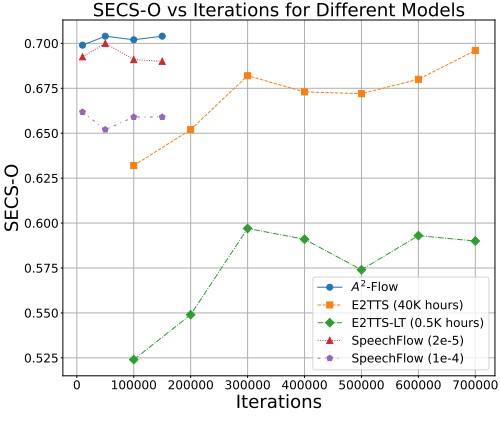

(a) WER vs Iterations for Different Models

(b) SECS-O vs Iterations for Different Models

Figure 2: Comparison of WER and SECS-O across training iterations for various models

pre-training followed by fine-tuning on the LibriTTS dataset, E2TTS-LT trained on LibriTTS, and E2TTS trained on 40K hours of transcribed data. Fig. 2 illustrates the pronunciation accuracy (WER) and speaker similarity (SECS-O) during zero-shot TTS for each model across different training iterations.

While E2TTS efficiently learns text-speech alignment with a large amount of transcribed data, it struggles under low-resource conditions, requiring over 400K iterations to achieve acceptable performance with limited data. In contrast, $A^2$-Flow, with its pre-trained alignment capabilities, achieves comparable results with significantly fewer fine-tuning iterations.

On the other hand, when SpeechFlow is pre-trained using only masked speech modeling and then fine-tuned on the LibriTTS dataset, it struggles to learn text-speech alignment. Although Speech-Flow achieves high speaker similarity when fine-tuned with learning rate of $2e^{-5}$, it fails to learn text-speech alignment, resulting in WER values exceeding 100%. Increasing the learning rate to $1e^{-4}$ leads to a drop in speaker similarity without effectively improving alignment.

The performance curve of E2TTS-LT further highlights the challenges of learning text-speech alignment from scratch using only a small amount of data, as it requires many iterations to converge. This demonstrates that SpeechFlow, when trained solely with masked speech modeling, is not an effective initialization for TTS models requiring joint text-speech alignment. In contrast, $A^2$-Flow's alignment-aware pre-training with discrete speech units makes it highly efficient for learning text-speech alignment during TTS fine-tuning, even with limited data.

## 4.2 MODEL COMPARISONS

Table 1: Objective metric results on the TTS Cross Reference Synthesis task using the LibriSpeech test-clean dataset. † indicates results directly reported by each model, while * represents results obtained from re-implemented experiments. "SpeechFlow-E2*" refers to the SpeechFlow model fine-tuned using the E2TTS approach. "PT" indicates whether pre-training was used.

| MODEL | PT | UNLABELED DATA (H) | LABELED DATA (H) | WER↓ | SECS-O↑ |
|---|---|---|---|---|---|
| CLAM-TTS[†] | ✗ | 0 | 55,000 | 5.11 | 0.495 |
| DiTTO-TTS[†] | ✗ | 0 | 55,000 | 2.56 | 0.627 |
| VOICEBOX[†] | ✗ | 0 | 60,000 | 1.9 | 0.662 |
| SPEECHFLOW[†] | ✓ | 60,000 | 960 | 2.1 | 0.700 |
| E2TTS[†] | ✗ | 0 | 50,000 | 2.0 | 0.675 |
| E2TTS [†] | ✓ | 200,000 | 50,000 | 1.9 | 0.708 |
| E2TTS* | ✗ | 0 | 40,000 | 1.9 | 0.696 |
| E2TTS-LT* | ✗ | 0 | 500 | 4.0 | 0.590 |
| SPEECHFLOW-E2* | ✓ | 40,000 | 500 | 107.7 | 0.690 |
| $A^2$-FLOW | ✓ | 40,000 | 500 | 2.2 | 0.704 |

**Zero-shot Text-to-Speech**  We compare the performance of $A^2$-Flow, fine-tuned solely on the LibriTTS dataset, with other zero-shot TTS models. Table 1 shows that $A^2$-Flow achieves comparable pronunciation accuracy (WER) and higher speaker similarity (SECS-O) than most baselines, including Voicebox. $A^2$-Flow also performs on par with E2TTS while using as little as 1% of the transcribed data required by most zero-shot TTS models.

Compared to E2TTS (Eskimez et al., 2024), which required 200K hours of unlabeled data and 800K fine-tuning iterations with 50K hours of labeled data to achieve a WER of 1.9 and SECS-O of 0.708, $A^2$-Flow achieves similar performance with significantly fewer fine-tuning iterations, far less labeled data, and reduced computational resources. In contrast, SpeechFlow, when fine-tuned using the E2TTS framework (referred to as SpeechFlow-E2), achieves a high SECS-O score but fails to achieve reasonable WER values, highlighting the difficulty of adapting SpeechFlow to tasks requiring text-speech alignment.

These results demonstrate that $A^2$-Flow provides an efficient alternative to existing pre-training methods, effectively modeling alignment while minimizing reliance on transcribed data and computational resources. By leveraging alignment-aware pre-training, $A^2$-Flow delivers robust zero-shot TTS performance, requiring only a fraction of the resources used by the previous approaches like E2TTS.

Table 2: A/B test results comparing $A^2$-Flow against SpeechFlow and E2TTS, respectively. "Win" indicates cases where $A^2$-Flow was preferred.

| MODEL | WIN-TIE-LOSE |
|---|---|
| $A^2$-FLOW VS SPEECHFLOW | $41.6\% - 24.4\% - 34.0\%$ |
| $A^2$-FLOW VS E2TTS | $34.8\% - 30.2\% - 35.0\%$ |

To further validate our model, we conducted a subjective A/B test to compare the samples generated by $A^2$-Flow against those from E2TTS and SpeechFlow. Evaluators were asked to select the sample that better matched the reference audio in terms of prosody, emotion, and timbre. The results of the A/B test, presented in Table 2, show that $A^2$-Flow is almost comparable to E2TTS, with a slight preference towards E2TTS, and outperforms SpeechFlow (Liu et al., 2024). The results show that $A^2$-Flow performs comparably to E2TTS and outperforms SpeechFlow in subjective evaluations, highlighting its ability to efficiently model text-speech alignment without relying on an external alignment mechanism. We have uploaded the samples used in the subjective evaluation to the demo page link in Section A.1, and we encourage readers to listen to the samples on the demo page.

Table 3: TTS Continuation results for non-English languages. Non-English models are trained on respective CML datasets and evaluated on their corresponding test sets.

| MODEL | LANGUAGE | FINE-TUNING DATASET | LABELED DATA (H) | WER↓ | SECS-O↑ |
|---|---|---|---|---|---|
| GT | GERMAN | – | – | 7.5 | 0.628 |
| $A^2$-FLOW | | CML-GERMAN | 1400 | 7.6 | 0.609 |
| GT | SPANISH | – | – | 5.1 | 0.674 |
| $A^2$-FLOW | | CML-SPANISH | 440 | 6.2 | 0.634 |
| GT | FRENCH | – | – | 6.0 | 0.619 |
| $A^2$-FLOW | | CML-FRENCH | 280 | 7.7 | 0.564 |

To show that $A^2$-Flow can achieve high zero-shot TTS performance across languages, we fine-tune the model on three languages from the CML-Dataset—German, Spanish, and French—using 150K iterations for each. We evaluate the models on the test sets of each language and perform the TTS continuation task for samples between 4 and 10 seconds, as done on LibriSpeech. As shown in Table 3, $A^2$-Flow effectively learns text-speech alignment for each language, demonstrating pronunciation accuracy (WER) and speaker similarity (SECS-O) that are not significantly worse compared to the ground truth, despite differences in language. These results indicate that $A^2$-Flow can leverage untranscribed data to build strong zero-shot TTS models for multiple languages, even when large-scale transcribed data is unavailable.

**Zero-shot Voice Conversion**  We compare our model with three voice conversion baselines—Any-to-Any VC (Kovela et al., 2023), UnitSpeech (Kim et al., 2023a), and SelfVC (Neekhara et al., 2024)—using the LibriSpeech test-clean dataset. Among these, Any-to-Any VC is a VC baseline directly trained on the test-clean dataset, while UnitSpeech is a fine-tuning-based VC baseline that

Table 4: Objective metric results for the Voice Conversion Cross Reference Synthesis task.

| MODEL | ZERO-SHOT | WER↓ | SECS-O↑ |
|---|---|---|---|
| ANY-TO-ANY VC | ✗ | 3.0 | 0.648 |
| UNITSPEECH | ✗ | 2.9 | 0.674 |
| SELFVC | ✓ | 3.2 | 0.375 |
| $A^2$-FLOW | ✓ | 3.6 | 0.67 |

adapts to the reference speech with 500 iterations of fine-tuning. SelfVC serves as a zero-shot VC baseline. The results of cross-reference synthesis for each model are presented in Table 4. Although $A^2$-Flow shows a slightly higher WER compared to other baselines, it achieves significantly higher speaker similarity than the zero-shot baseline SelfVC. Additionally, it demonstrates performance comparable to or exceeding that of baselines directly trained or fine-tuned on the test set's reference audio, highlighting the effectiveness of large-scale unit-based alignment-aware pre-training for voice conversion.

## 4.3 ABLATION STUDY

We perform an ablation study on the pre-trained $A^2$-Flow model for TTS tasks, exploring the impact of different sampling steps, classifier-free guidance scales ($\gamma$), and timestep shifting scales ($\alpha$) on cross-reference synthesis performance. Table 5 shows that setting classifier-free guidance scale $\gamma = 1$ results in significantly worse WER and SECS-O, confirming $\gamma = 2$ as the optimal value. A key finding of this work is that, for a flow matching model trained to jointly model alignment in TTS, using timestep shifting with $\alpha = 3$, which samples more frequently from the noisier $t$ regions, results in improved pronunciation accuracy and better speaker similarity compared to uniform timestep sampling ($\alpha = 1$) at the same number of sampling iterations. Therefore, we use $\alpha = 3$ as the default value. For Euler sampling steps, performance does not improve beyond 32 steps, while reducing steps to 16 leads to a drop in performance. Thus, we use 32 steps as the default setting.

Table 5: Ablation study results of $A^2$-Flow on zero-shot TTS with variations in Euler sampling steps, classifier-free guidance scale $\gamma$, and timestep shifting scale $\alpha$.

| MODEL | NFE | CFG SCALE | $\alpha$ | WER↓ | SECS-O↑ |
|---|---|---|---|---|---|
| | | | 1 | 2.7 | 0.695 |
| | 32 | 2 | 2 | 2.3 | 0.703 |
| | | | 3 | 2.2 | 0.704 |
| | | 1 | | 3.3 | 0.679 |
| $A^2$-FLOW | 32 | 2 | 3 | 2.2 | 0.704 |
| | | 3 | | 2.3 | 0.706 |
| | 16 | | | 2.7 | 0.698 |
| | 32 | 2 | 3 | 2.2 | 0.704 |
| | 64 | | | 2.3 | 0.701 |

## 5 CONCLUSION

In this work, we proposed $A^2$-Flow, an alignment-aware pre-training approach tailored for speech synthesis tasks that involve generating natural phonetic content, such as voice conversion and text-to-speech (TTS). By incorporating de-duplicated unit sequences instead of text into the E2TTS framework, our method enables the model to learn the alignment between input units and speech during the pre-training phase. This alignment-aware pre-training can be directly applied to zero-shot voice conversion tasks or used to build a TTS model that jointly models text-speech alignment with minimal fine-tuning. As a result, $A^2$-Flow achieves comparable performance to state-of-the-art models like E2TTS using only 1% of the transcribed data. Moreover, our method consistently outperforms existing zero-shot TTS and zero-shot VC models by a significant margin. We further demonstrated that $A^2$-Flow can model text-speech alignment for multiple languages, making it adaptable to multilingual TTS scenarios. Our findings highlight that $A^2$-Flow is better suited for alignment-aware tasks compared to pre-training methods like SpeechFlow, which do not incorporate any conditioning. Since our approach can learn effectively with only speech data and language IDs, $A^2$-Flow offers a viable solution for scenarios where large-scale transcribed datasets are not available, and provides significant advantages for alignment-aware speech synthesis tasks.

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

# A   APPENDIX

## A.1   DEMO PAGE

The demo page link is https://anonymous.4open.science/r/demo-page-B24F/index.md. The demo page provides the comparison samples used for subjective evaluation as well as the samples generated by the models used for voice conversion.

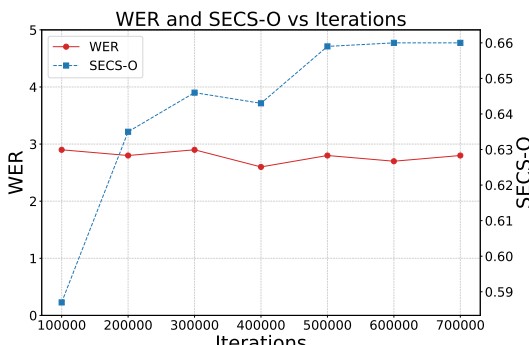

Figure 3: Reconstruction performance of A$^2$-Flow during pre-training across different training iterations.

## A.2   MODEL DETAILS

### A.2.1   TOTAL LENGTH PREDICTOR

A$^2$-Flow utilizes a total length predictor to estimate the duration of the speech corresponding to the input text sequence. The total length predictor is composed of a convolution-based pre-network for projecting the reference audio and a transformer network for predicting the total length from the reference audio and the text input. Specifically, the total length predictor takes the reference audio $x_{\text{ref}}$ and the text input sequence $y$ as inputs. The reference audio $x_{\text{ref}}$ is first processed through the convolutional pre-network, which consists of three convolutional layers, each with a kernel size of 5, to project $x_{\text{ref}}$ into an intermediate representation $h_{\text{ref}}$. Then, a special token $[SPC]$ is prepended to the text input sequence $y$ to provide a placeholder for predicting the total length. The representations $h_{\text{ref}}$, the special token $[SPC]$, and the text input sequence $y$ are concatenated along the time axis and input into the transformer network.

The transformer network employs Rotary Positional Embedding (RoPE) to capture positional information, with separate rotary embeddings applied to the reference audio and text input sequence. The network is configured with a hidden size of 512, 8 layers, and 8 heads in the multi-head attention mechanism, resulting in a total of 30 million parameters. The output from the transformer network is taken from the position of the special token $[SPC]$, which is then projected into a scalar value. This scalar represents the logarithmic value of the speech length $d$, scaled by a scaling factor $s$, i.e., $\log \frac{d}{s}$. This design enables the total length predictor to effectively estimate the duration of the generated speech while leveraging both reference audio and text input.

### A.2.2   ADDITIONAL EXPLANATION ON TIMESTEP SHIFTING

A$^2$-Flow utilizes the timestep shifting function in Eq. 7 for sampling during the inference process. Fig. 4 illustrates the timestep shifting function depending on the value of the timestep shifting scale $\alpha$. When $\alpha = 1$, the function performs uniform sampling, similar to the standard Euler method. As the shifting scale $\alpha$ increases, more timesteps are sampled near the noise-dominated region ($t = 0$). This allows for finding more accurate alignment between the input and the speech during smaller timesteps.

The sampling process is shown in Alg. 1. During sampling, as described in Section 2, the uniformly sampled timesteps $t_i = \frac{i}{N}$ are mapped to $\hat{t}_i$ using the timestep shifting function. The ODE is then solved iteratively at these shifted timesteps to generate samples. At each timestep of the ODE

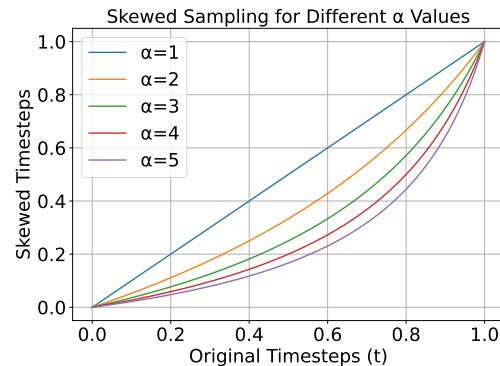

Figure 4: Visualization of the timestep shifting function according to the value of $\alpha$.

solution, the algorithm performs repetitive computations for the vector field value, depending on whether classifier-free guidance is applied.

---

**Algorithm 1** Skewed Sampling for Solving ODE

---

1: Initialize $x_0 \sim N(O, I)^T$
2: Set the following parameters:
3:    $c$ = all conditioning inputs
4:    $N$ = number of sampling steps
5:    $\alpha$ = timestep shifting scale
6:    $\gamma$ = classifier-free guidance scale
7: Define $f(t) = \frac{t}{1+(\alpha-1)(1-t)}$    // Timestep shifting function
8: **for** $i = 0$ to $N - 1$ **do**
9:    $t_i \leftarrow \frac{i}{N}$, $t_{i+1} \leftarrow \frac{i+1}{N}$
10:    $\hat{t}_i \leftarrow f(t_i)$, $\hat{t}_{i+1} \leftarrow f(t_{i+1})$
11:    $v_c \leftarrow v(x_{\hat{t}_i}, c, \hat{t}_i; \theta)$    // Conditional vector field
12:    $v_u \leftarrow v(x_{\hat{t}_i}, \phi, \hat{t}_i; \theta)$    // Unconditional vector field
13:    $v_{\hat{t}_i} \leftarrow v_c + \gamma \cdot (v_c - v_u)$
14:    $x_{\hat{t}_{i+1}} \leftarrow x_{\hat{t}_i} + v_{\hat{t}_i} \cdot (\hat{t}_{i+1} - \hat{t}_i)$
15: **end for**
16: **return** $x_1$

---

### A.3 ADDITIONAL RESULTS

Table 6: Objective metric and UTMOS results on the TTS Cross Reference Synthesis task using the LibriSpeech test-clean dataset. $*$ represents results obtained from re-implemented experiments. "SpeechFlow-E2*" refers to the SpeechFlow model fine-tuned using the E2TTS approach. "PT" indicates whether pre-training was used.

| MODEL | PT | UNLABELED (H) | LABELED (H) | WER↓ | SECS-O↑ | UTMOS↑ |
|-------|----|--------------|-------------|------|---------|--------|
| E2TTS* | ✗ | 0 | 40,000 | 1.9 | 0.696 | 4.01 |
| E2TTS-LT* | ✗ | 0 | 500 | 4.0 | 0.590 | 4.01 |
| SPEECHFLOW-E2* | ✓ | 40,000 | 500 | 107.7 | 0.690 | 3.68 |
| A$^2$-FLOW | ✓ | 40,000 | 500 | 2.2 | 0.704 | 4.03 |

**Speech Naturalness** We present the naturalness evaluation results for samples generated by the reproduced baselines E2TTS, E2TTS-LT, SpeechFlow-E2, and A$^2$-Flow on the LibriSpeech test-clean dataset in Table 1. To assess the naturalness of the samples, we use UTMOS (Saeki et al., 2022), a model trained to predict the Mean Opinion Score (MOS) for audio naturalness, and report the average predicted MOS as a proxy for naturalness. Table 6 shows the UTMOS scores for each

model, with the ground truth audio of all evaluation samples achieving an average UTMOS score of 4.10. In comparison, both the reproduced E2TTS and our $A^2$-Flow demonstrate the ability to generate highly natural audio.

**Performance Comparison of E2TTS and $A^2$-Flow**  We compare the performance of E2TTS and $A^2$-Flow across different training iterations. For E2TTS, we evaluate both the original model trained for the total number of iterations used in previous comparisons and an extended version trained for an additional 150K iterations, resulting in a total of 850K iterations. For $A^2$-Flow, we present results fine-tuned solely on LibriTTS for 150K and 300K iterations. Additionally, we evaluate the performance of "$A^2$-Flow-T," which was fine-tuned on the entire 40K hours of labeled data, similar to E2TTS.

In Table 1, we compared the reproduced results of $A^2$-Flow and E2TTS using only the proposed sampling method, timestep shifting, during inference. Here, we also provide performance results without timestep shifting, where the timestep shifting scale is set to $\alpha = 1$.

Table 7 summarizes the results of E2TTS and $A^2$-Flow across different training setups, using $\alpha = 3$ as the timestep shifting scale during inference. All E2TTS models were trained using 32 GPUs, while $A^2$-Flow utilized 32 GPUs for the full pre-training phase and fine-tuning on the same amount of transcribed data as E2TTS. Fine-tuning on LibriTTS alone was performed with 8 GPUs.

The experimental results show that E2TTS exhibits minimal improvement in objective metrics even after 150K additional iterations beyond the initial 700K. In contrast, $A^2$-Flow achieves a higher SECS-O score with just 150K iterations of fine-tuning. When fine-tuned on LibriTTS for 300K iterations, $A^2$-Flow slightly sacrifices SECS-O for improved WER, achieving results comparable to E2TTS. Furthermore, $A^2$-Flow-T, fine-tuned on 40K hours of transcribed data like E2TTS, maintains a similar WER while pushing SECS-O to 0.711, demonstrating the effectiveness of alignment-aware pre-training even with the same amount of transcribed data.

For E2TTS without timestep shifting ($\alpha = 1$), the results show a slight degradation in WER compared to its counterpart using timestep shifting ($\alpha = 3$). Overall, these results highlight that the proposed timestep shifting method improves alignment accuracy for both $A^2$-Flow and E2TTS, further validating its utility.

Table 7: Objective metric results of E2TTS and $A^2$-Flow across different training iterations and timestep shifting scales on the LibriSpeech test-clean dataset. * represents results obtained from re-implemented experiments. Training details specify the number of iterations and GPU configurations used.

| MODEL | TRAINING DETAILS | $\alpha$ | WER↓ | SECS-O↑ | UTMOS↑ |
|---|---|---|---|---|---|
| E2TTS* | 700K (32GPU) | 3 | 1.9 | 0.696 | 4.01 |
| E2TTS* | 850K (32GPU) | 3 | 2.0 | 0.695 | 4.02 |
| E2TTS* | 700K (32GPU) | 1 | 2.1 | 0.697 | 3.98 |
| E2TTS* | 850K (32GPU) | 1 | 2.2 | 0.690 | 4.01 |
| $A^2$-FLOW | 700K (32GPU) / 150K (8GPU) | 3 | 2.2 | 0.704 | 4.03 |
| $A^2$-FLOW | 700K (32GPU) / 300K (8GPU) | 3 | 1.9 | 0.695 | 4.06 |
| $A^2$-FLOW-T | 700K (32GPU) / 150K (32GPU) | 3 | 2.0 | 0.711 | 4.01 |
| $A^2$-FLOW | 700K (32GPU) / 150K (8GPU) | 1 | 2.6 | 0.699 | 4.00 |
| $A^2$-FLOW | 700K (32GPU) / 300K (8GPU) | 1 | 2.1 | 0.690 | 4.04 |
| $A^2$-FLOW-T | 700K (32GPU) / 150K (32GPU) | 1 | 2.2 | 0.710 | 3.99 |

**Pre-training with Different Units**  In our experiments, we utilized the HuBERT expresso model [2], pre-trained on 220K hours of data, for alignment-aware pre-training. To investigate the impact of different HuBERT unit representations on TTS performance, we provide results using an alternative HuBERT model. Specifically, we use the HuBERT-base model trained on 960 hours of LibriSpeech, employing units with a clustering size of $K = 200$. We pre-train the model on the same 40K hours of unlabeled data and fine-tune it on 500 hours of the LibriTTS dataset to compare results.

Table 8 presents the TTS downstream performance with different unit representations used during pre-training. Our results demonstrate that the proposed alignment-aware pre-training method

---

[2]https://github.com/facebookresearch/textlesslib/tree/main/examples/expresso

achieves comparable performance even when using HuBERT units trained with minimal data and only 200 clusters. This highlights the robustness of our pre-training approach with respect to the choice of unit representation.

Table 8: Objective metric results for A$^2$-Flow on the TTS Cross Reference Synthesis task using the LibriSpeech test-clean dataset with different HuBERT unit representations during pre-training. HuBERT-Expresso (K=2000) refers to the model pre-trained on 220K hours of data, while HuBERT-LS960 (K=200) refers to the model trained on 960 hours of LibriSpeech.

| MODEL | HuBERT UNITS | WER↓ | SECS-O↑ | UTMOS↑ |
|---|---|---|---|---|
| A$^2$-FLOW | HuBERT-EXPRESSO (K=2000) | 2.2 | 0.704 | 4.03 |
| A$^2$-FLOW | HuBERT-LS960 (K=200) | 2.2 | 0.703 | 4.02 |

### A.4 LIMITATION

A limitation of A$^2$-Flow is that it relies on self-supervised speech units for pre-training, making it less generalizable to non-speech audio domains compared to methods like SpeechFlow, which do not use any specific conditioning. Another limitation arises from the E2TTS framework itself, which necessitates a separate total length predictor to estimate the overall speech duration, preventing joint modeling of the total duration of generated speech. Investigating pre-training methods that can jointly model all aspects of speech generation including total length of the audio could be a promising direction for future research.

### A.5 HUMAN EVALUATION METHOD

To compare the performance of A$^2$-Flow with other models, we used the Defined AI [3] platform to conduct human evaluations. Each evaluation judgment was compensated at a rate of $0.15, and the evaluations were performed by 100 human evaluators, all located in the United States. Evaluators were selected based on the platform's internal agreement score filtering criteria to ensure reliability.

Each evaluator assessed all provided samples, with payments calculated based on the number of judgments completed. Specifically, evaluators were compensated for 19 judgments for comparisons with E2TTS and 9 judgments for comparisons with SpeechFlow. During the evaluation, each evaluator was presented with a 3-second reference audio clip alongside two audio samples generated by the models in a randomized order. The following instruction was provided:

"Between the two samples, which one sounds closer to the reference in terms of prosody, emotion, and timbre? If the two samples sound equally similar to the reference, choose 'Neither.'"

This setup encouraged evaluators to consider the overall quality of the generated audio in relation to the reference audio, encompassing aspects such as prosody, emotion, and timbre, to determine which sample was better.

---

[3]https://defined.ai/crowd-as-a-service

