# OpenReview forum: "A$^2$-Flow: Alignment-Aware Pre-training for Speech Synthesis with Flow Matching"
_ICLR.cc/2025/Conference — Submitted to ICLR 2025_

### Official Review · Reviewer_DdFz · 2024-10-31

**Soundness:** 2
**Presentation:** 2
**Contribution:** 2
**Rating:** 3
**Confidence:** 5

**Summary:**

The paper introduces A2-Flow, a alignment-aware pre-training method for flow matching models in speech synthesis. This approach facilitates alignment-free voice conversion and enhances convergence speed during TTS fine-tuning, making it particularly effective in low-resource settings.

**Strengths:**

1. Good presentation.
2. One framework can solve VC and TTS tasks.

**Weaknesses:**

1. It is hard to understand what alignment-aware pre-training is and its advanteges.

2. It is difficult to grasp the difference between using de-duplicated HuBERT tokens and text for pre-training. In my view, the only advantage is support for voice conversion tasks. However, the idea of a single model addressing multiple audio tasks has been widely discussed in previous works, such as VoiceBox, AudioBox. Thus, it is challenging to view this as a novel contribution.

3. For the TTS task, this work still requires a sentence duration predictor to control duration. Such strategies have been widely used in diffusion-based TTS systems, including SimpleSpeech, DiTTo-TTS, and E3TTS, and so on.

I am happy to discuss this paper further with the authors and other reviewers during the discussion stage.

**Questions:**

(1) Please explain the advanteges of alignment-aware pre-training?

(2) Can you summarize your core contributions? For example, list them as 1,2,3... From this version, I cannot understand your core contributions.

(3) VC task does not need any alignment information, because the output duration can always the same as the input audio. Your TTS model still need additional sentence duration prediction model. So, why do you propose alignment-aware pre-training?

---

> ### Author Response · Authors · 2024-11-28
>
> ### Weakness 1 & Q1. **Regarding Alignment-aware pre-training**
>
> - **Importance of alignment modeling** First, discussing the importance of alignment modeling, effective alignment modeling is crucial in the TTS field. Non-autoregressive models, from FastSpeech and Glow-TTS to the most recent VoiceBox, have explicitly modeled phoneme durations using a separate duration predictor. In these models, a phoneme duration predictor predicts the duration for each phoneme, expands the phoneme sequence based on the predicted durations, pre-aligns it with speech, and generates audio from the aligned sequence using a decoder. However, as explained in the official comment, the E2TTS approach, which inherently aligns input text tokens with speech through self-attention without using a separate text-token-based duration predictor, has been shown in the E2TTS paper to outperform models like VoiceBox.
>
>   - Implicit alignment modeling holds greater potential in terms of performance in TTS field.
>
> - Existing models like E2TTS and DiTTO-TTS, which learn alignment jointly, have relied on large amounts of transcribed data for training. However, as shown in Figure 2 of this paper, E2TTS-LT, which uses a smaller amount of transcribed data, significantly underperforms compared to E2TTS in zero-shot TTS performance.
>
>   - Zero-shot TTS models that internally perform alignment modeling typically require a large amount of transcribed data.
>
> - The alignment-aware pre-training method proposed in this paper is designed for generative models like E2TTS that inherently align text sequences with speech. The key idea, inspired by E2TTS, is to train the model to jointly align a de-duplicated unit sequence extracted from unlabeled speech with the speech itself and generate speech. **During the fine-tuning process, the unit sequence is replaced with text input, allowing the model to adapt the ability to align unit sequences with speech during pre-training into aligning text with speech.**
>
> - This approach enables the construction of models like E2TTS, allowing the model to jointly align text and speech while generating audio with relatively little transcribed data and minimal fine-tuning. Additionally, unlike traditional TTS models, the proposed method offers the added advantage of supporting voice conversion.
>
> - Pre-training methods like SpeechFlow work well with approaches like VoiceBox, which rely on an explicit phoneme duration predictor, but as shown in Figure 2 of this paper, they are not well-suited as initialization for models like E2TTS that jointly perform internal alignment modeling. This highlights the advantages of the proposed method as a pre-training strategy for models that inherently learn alignment.
>
> **In summary, E2TTS has recently achieved the best performance in the zero-shot TTS field compared to methods using explicit duration predictors. This paper proposes a pre-training method that enables such a model to be built using as little as 1% of transcribed data, making it significantly more efficient.**
>
> ---
>
> ### Weakness 2.
>
> The ability to perform voice conversion is an additional advantage, but this method does not focus on handling multiple tasks. As mentioned earlier, the primary contribution of the paper is proposing a pre-training method that introduces unit representations from unlabeled data to enable alignment learning within the E2TTS framework. This approach minimizes reliance on transcribed data while building speech generative models like E2TTS capable of alignment modeling. As a note, both VoiceBox and AudioBox require explicit duration predictors and do not perform internal alignment modeling.
>
> ### Weakness 3.
>
> Using a sentence duration predictor is not our main contribution. The models you mentioned, such as SimpleSpeech, DiTTO-TTS, and E3TTS, all perform implicit alignment modeling, similar to E2TTS. However, the E2TTS model, which is primarily used in our paper, demonstrates significantly better performance compared to these approaches. This paper shows that we can achieve performance comparable to E2TTS using only 1% of transcribed data, leveraging unlabeled speech for alignment-aware pre-training.
>
> ### Q2.
>
> We provide our core contributions in the official comment above.
>
> ### Q3.
>
> When we refer to alignment modeling, we mean the alignment between the input sequence and speech. In TTS models that use explicit duration predictors, the role of the duration predictor, which models phoneme durations representing the alignment, is crucial. In contrast, our model internally models this alignment information. While it uses a sentence duration predictor, it inherently models the alignment between unaligned units or text tokens and speech. This is why we call it alignment-aware pre-training. Additionally, in the VC task, since de-duplicated units are used as input, the model can internally align the input source speaker and generate modulated speech with a different alignment, offering an additional advantage.

---

> > ### Comment · Reviewer_DdFz · 2024-11-28
> >
> > Thank you for your response.
> >
> > I understand that the proposed Alignment-aware pre-training method is similar to E2TTS, as it replaces the character sequence with a de-duplicated semantic sequence. However, I have the following concerns:
> >
> > **Limited Novelty:** The pre-training paradigm is adopted from E2TTS. The paper emphasizes the importance of unlabeled speech data pre-training, but it still requires additional pre-trained SSL models (such as HuBERT). Previous works, such as SimpleSpeech [1], also use unlabeled speech, but they rely on pre-trained Whisper for transcription. The main difference between your method and theirs is the use of HuBERT or Whisper. From my perspective, the two approaches are very similar, and it’s hard to determine which one is better.
> >
> > "Implicit alignment modeling holds greater potential in terms of performance in the TTS field." How do you support this claim? From your experimental results, the robustness of your model still seems poorer than previous works, such as E2TTS and NaturalSpeech 3. I agree that implicit alignment could bring advantages, like allowing the use of more speech data, but at this stage, these models still perform worse than force-alignment-based TTS (e.g., NaturalSpeech 3).
> >
> > Pre-training and Fine-tuning with Different Inputs: The paper shows that using different types of input (semantic tokens during pre-training and text sequences during fine-tuning) performs worse than using only text sequences (as in E2TTS).
> >
> > Sentence Duration Prediction: The sentence duration prediction significantly influences the quality of the synthesized speech. Did the authors compare using ground truth (GT) sentence duration versus predicted sentence duration?
> >
> > "Using a sentence duration predictor is not our main contribution." I agree with this statement. Additionally, as I mentioned earlier, learning alignment internally within TTS models is not a new idea. Works like SimpleSpeech [1], E2TTS [2], and DiTTO-TTS [3] have already explored this approach. Replacing the text sequence with a semantic sequence cannot be considered a novel contribution, especially considering the standards of ICLR.
> >
> > In summary, I stand by my score. The novelty of the method and the experimental results are insufficient to justify publication at ICLR.
> >
> > [1] Dongchao Yang, Dingdong Wang, Haohan Guo, Xueyuan Chen, Xixin Wu, Helen Meng, "SimpleSpeech: Towards Simple and Efficient Text-to-Speech with Scalar Latent Transformer Diffusion Models", Interspeech 2024.
> >
> > [2] Eskimez S E, Wang X, Thakker M, et al. E2 tts: Embarrassingly easy fully non-autoregressive zero-shot tts[J]. SLT, 2024
> >
> > [3] Lee K, Kim D W, Kim J, et al. DiTTo-TTS: Efficient and Scalable Zero-Shot Text-to-Speech with Diffusion Transformer[J]. arXiv preprint arXiv:2406.11427, 2024.

---

> > > ### Author Response · Authors · 2024-11-29
> > >
> > > **Limited Novelty**
> > >
> > > A²-Flow, methodologically, involves pretraining with <de-duplicated unit, speech> pairs and fine-tuning with <text, speech> pairs. However, from this approach, we demonstrated that “fine-tuning a model pretrained with <unit, speech> pairs leads to effective <text, speech> alignment learning.” This allowed us to achieve state-of-the-art zero-shot TTS performance with a small amount of transcribed data and showed that utilizing units also achieves strong performance in zero-shot voice conversion (VC).
> > >
> > > The reviewer seems to view our approach as methodologically unoriginal, perceiving it as merely replacing the text sequence in E2TTS with a de-duplicated unit sequence. However, the novelty of the paper lies not in the methodological simplicity but in the achievements it brings and the intuitions it provides.
> > >
> > > For example, the pretraining method of SpeechFlow, while simple in that it removes the aligned phonemes from VoiceBox, demonstrated its utility for speech generative pretraining. Similarly, E2TTS replaced VoiceBox's aligned phonemes with unaligned character sequences and showed that this straightforward idea could simultaneously learn alignment and outperform the duration-predictor-based approach of VoiceBox, achieving state-of-the-art performance. Although these methodologies are simple, the impact of the results they achieved is significant.
> > >
> > > Similarly, while our method may appear simple, we demonstrate that the alignment learning capability within E2TTS is transferable. It is not an obvious approach that fine-tuning the model pretrained with <unit, speech> pairs would naturally lead to effective <text, speech> alignment learning. For example, in both Figure 2 (SpeechFlow) and E2TTS-LT, alignment learning using the E2TTS framework with limited transcribed data required over 400K iterations in the case of E2TTS-LT before the WER dropped below 10. Furthermore, SpeechFlow failed entirely to achieve alignment learning when fine-tuned with the same amount of data as A²-Flow. In contrast, A²-Flow efficiently learns alignment between text and speech during a small amount of additional fine-tuning. This demonstrates that alignment learning between <unit, speech> significantly benefits <text, speech> alignment learning.
> > >
> > > Additionally, methodologically, it is not trivial that we do not simply use the 50Hz unit sequence from HuBERT but instead propose a pretraining approach capable of learning alignment internally. By removing repetitions in the unit sequence to create de-duplicated units, we effectively guide alignment learning.
> > >
> > > To our knowledge, no prior research has demonstrated this, making it a non-trivial idea. Furthermore, we are the first to show the potential of distilling alignment capabilities, achieving state-of-the-art zero-shot TTS performance while requiring only a minimal amount of transcribed data.
> > >
> > > ---
> > >
> > > **(Regarding SimpleSpeech)** SimpleSpeech is a model that trains TTS using ASR-transcribed data extracted from unlabeled data via Whisper and performs zero-shot TTS using a speaker encoder. To compare with SimpleSpeech, we evaluated the zero-shot TTS performance of A²-Flow on 20 samples from the SimpleSpeech demo page ([prompt link](https://github.com/simplespeech/simplespeechDemo/tree/main/prompt2), [simplespeech link](https://github.com/simplespeech/simplespeechDemo/tree/main/ours)).
> > >
> > > Using the WavLM-TDNN speaker encoder employed in our paper to measure SECS-O, we computed the average SECS-O for these 20 samples as follows:
> > > | Model | SECS-O |
> > > |--|--|
> > > | A²-Flow| 0.76|
> > > | SimpleSpeech| 0.36|
> > >
> > > These results indicate that SimpleSpeech performs significantly worse than A²-Flow in zero-shot speaker adaptation. In addition, we demonstrated in our paper that A²-Flow surpasses various existing zero-shot TTS methods (e.g., VoiceBox, DiTTO-TTS) and is comparable to the state-of-the-art method E2TTS. Moreover, we show that alignment capability is transferable, which is not addressed by SimpleSpeech.
> > >
> > > Furthermore, not only SimpleSpeech but also other zero-shot TTS models trained using ASR-transcribed data, such as the E2TTS trained in F5-TTS as mentioned by Reviewer staD, relied on heavily filtered data using language ID, speech quality, average character duration, and other criteria. Despite this extensive filtering, the trained E2TTS achieved a high speaker similarity with an SECS-O of 0.69 but suffered from poor pronunciation accuracy with a WER of 2.9. This highlights the critical dependency of such methods on the accuracy of the ASR model and the filtering process.
> > >
> > > In contrast, our method does not require accurate ASR transcripts; it can directly utilize the unit sequence generated from HuBERT, emphasizing its simplicity and ease of use.

---

> > > > ### Author Response · Authors · 2024-11-29
> > > >
> > > > **Regarding Comparison w/ baselines**
> > > > I am unsure why the reviewer considers our proposed method to underperform compared to E2TTS and NaturalSpeech 3. As shown in Tables 1 and 7 of the paper, our approach achieves similar or even higher speaker similarity (SECS-O) than E2TTS using significantly less transcribed data. Could you clarify why our method is considered less effective? If it is due to WER, please note that the WER of the ground truth for all samples is 2.3.
> > > >
> > > > Additionally, in Table 7, we provided results for fine-tuning with the same amount of transcribed data as E2TTS (A²-Flow-T), showing an SECS-O of 0.711, which is higher than any previously published zero-shot TTS methods. Regarding NaturalSpeech 3, while direct numerical comparison is difficult since it evaluates one randomly sampled utterance per speaker in the test set instead of the entire sample set, the E2TTS paper, which shares authors with NaturalSpeech 3, reports a WER of 2.6 and SECS-O of 0.632.
> > > >
> > > > We recap that our pretraining followed by fine-tuning approach achieves comparable performance to E2TTS using less transcribed data and outperforms it (A²-Flow-T) when using the same amount of transcribed data.
> > > >
> > > > ### Table: Performance Comparison
> > > >
> > > > | Model                                   | WER  | SECS-O | UTMOS |
> > > > |-----------------------------------------|-------|---------|-------|
> > > > | Ground Truth                            | 2.3   | 0.700   | 4.10  |
> > > > | A²-Flow (duration predictor)            | 2.2   | 0.704   | 4.03  |
> > > > | A²-Flow-T (duration predictor)          | 2.0   | 0.711   | 4.01  |
> > > > | A²-Flow (GT duration)                   | 2.3   | 0.705   | 4.05  |
> > > > | E2TTS (700K iter, w/ our timestep shifting, α=3) | 1.9   | 0.696   | 4.01  |
> > > > | E2TTS (850K iter, w/ our timestep shifting, α=3) | 2.0   | 0.695   | 4.02  |
> > > > | E2TTS (700K iter, w/o our timestep shifting) | 2.1   | 0.697   | 3.98  |
> > > > | E2TTS (850K iter, w/o our timestep shifting) | 2.2   | 0.690   | 4.01  |
> > > > | E2TTS (official)                        | 2.0   | 0.675   | -     |
> > > > | VoiceBox                                | 1.9   | 0.662   | -     |
> > > > | DiTTO-TTS                               | 2.6   | 0.627   | -     |
> > > >
> > > > ---
> > > >
> > > > In response to Reviewer 98gk's request, we conducted experiments to evaluate the robustness of our total length predictor. The mean and 95% confidence interval of the ratio between the predicted length by our total length predictor and the ground truth length are as follows, showing that the ratio is generally close to 1 and well-predicted.
> > > >
> > > > | Duration (seconds) | Mean Ratio | 95% Confidence Interval |
> > > > |---------------------|------------|--------------------------|
> > > > | 3–4 seconds         | 1.119      | ± 0.015                 |
> > > > | 4–10 seconds        | 1.018      | ± 0.009                 |
> > > > | 10–15 seconds       | 1.021      | ± 0.014                 |
> > > >
> > > > When the ground truth sentence duration is used, as indicated in the table above, the WER and SECS-O are at a similar level to A²-Flow (GT duration). This demonstrates the robustness of our total length predictor.
> > > >
> > > > **Please consider the various advantages of our method mentioned above once again.**

---

> > > > ### Comment · Reviewer_DdFz · 2024-11-29
> > > >
> > > > Thank you for your response.
> > > >
> > > > From Table 1 in the paper, the best performance of E2TTS shows a WER of 1.9, while A2Flow achieves 2.2. This indicates that E2TTS outperforms A2Flow. Although the speaker similarity score is 0.704 for E2TTS and 0.696 for A2Flow, the gap is negligible and can be ignored. In fact, due to the gap between HuBERT tokens and text sequences, the robustness of A2Flow is likely to be worse than that of E2TTS.
> > > >
> > > > I understand the authors' view on the novelty of their work. The novelty may come from an engineering perspective. However, from a research perspective, the paper does not provide significant new insights for speech generation or machine learning. If the authors are willing to open-source their checkpoints, it could be a valuable engineering contribution to the community. But at this stage, I do not see a substantial contribution in the paper.
> > > >
> > > > Comparing this work to SimpleSpeech with SECS-O is unfair, as SimpleSpeech was only trained on 4K hours of data. Scaling is a crucial factor in achieving a high SECS-O score, and this comparison does not provide much insight. A better approach to highlight the advantages of your method would be to use the same dataset, combine it with the Whisper model for transcription, and then train E2TTS with this data. Without such an ablation study, it is difficult to conclude whether using SSL tokens during pre-training is a good strategy.
> > > >
> > > > Regarding the NaturalSpeech 3 baseline, their original paper reports a WER of 1.8. However, I noticed that they used a different test set from the one in your evaluation, which makes it difficult to directly compare A2Flow to their results.
> > > >
> > > > For duration prediction, it seems that your duration predictor is quite robust. However, your experimental results seem to contradict the findings in MASKGCT (https://arxiv.org/pdf/2409.00750).
> > > >
> > > > Lastly, although the authors claim that their model performs well, I did not see any generated samples in the paper. It is difficult to evaluate the performance of the model based solely on WER and SECS-O scores.
> > > >
> > > > In summary, I stand by my score. The incremental experiments on E2TTS are not sufficient for publication at ICLR.

---

> > > > > ### Author Response · Authors · 2024-12-03
> > > > >
> > > > > It seems that our contributions are being misrepresented. To clarify, we will unify the framework under the term **E2TTS** and consolidate the experimental results presented in our paper with those from the official E2TTS paper and the reproduced results of E2TTS in F5-TTS. Specifically, we will update Table 1, replacing the term "Speech-Flow-E2" with **E2TTS (SpeechFlow pretraining)**, and provide a comprehensive summary of the entire setup.
> > > > >
> > > > > ---
> > > > >
> > > > > | Setup| Details                  | Pre-training | Unlabeled | Labeled | WER | SECS-O | Notes|
> > > > > |-----|--|--|--|--|-----|--------|--|
> > > > > | **0)** Ground Truth |-|-|-|-| 2.3 | 0.700  ||
> > > > > | **1)** E2TTS (F5TTS re-implementation)|                            | X                     | 0       | 100,000   | 2.95| 0.690  | Used Whisper-transcribed data (Emilia dataset)                                           |
> > > > > | **2)** E2TTS (Official)               |           | ✓                     | 200,000 | 50,000    | 1.9 | 0.708  |                            |
> > > > > | **3)** E2TTS (Official)               |             | X                     | 0       | 50,000    | 2.0 | 0.675  |                                                                                           |
> > > > > | **4)** E2TTS (Our re-implementation)  |             | X                     | 0       | 40,000    | 1.9 | 0.696  |                                                                                           |
> > > > > | **5)** E2TTS (SpeechFlow pre-training)| 700K PT (32GPU) / 150K FT (8GPU)| ✓                  | 40,000  | 500       | 107 | 0.690  |                                                                                           |
> > > > > | **6)** E2TTS (A²-FLOW pre-training)   | 700K PT (32GPU) / 150K FT (8GPU)| ✓                  | 40,000  | 500       | 2.2 | 0.704  | Our proposed approach with better convergence rates                                       |
> > > > > | **7)** E2TTS (A²-FLOW pre-training)   | 700K PT (32GPU) / 150K FT (32GPU)| ✓                  | 40,000  | 40,000    | 2.0 | 0.711  | Finetuned on same amount (40K) of transcribed data |
> > > > >
> > > > > ---
> > > > > Our goal is to emphasize that our proposed pretraining method is more efficient than any existing approach, achieving state-of-the-art (SOTA) results while using significantly less data. By simulating a low-resource training scenario with only 500 hours of transcribed data, we demonstrate that our method achieves both efficiency and performance, requiring only **1/5** of the pretraining data and **1/100** of the transcribed data compared to other methods. Furthermore, we reiterate that the performance of E2TTS trained on Whisper-transcribed data is suboptimal, highlighting the robustness of our approach even under constrained data conditions.
> > > > >
> > > > > - **0)** Establishes the upper bound or expected target WER and SECS-O.
> > > > > - **1)** Shows that training with transcriptions obtained from Whisper yields poor WER, even after applying multiple filtering strategies.
> > > > > - **2)** Demonstrates that pretraining with inpainting (without alignment learning during pretraining) yields good SECS-O but requires large amounts of transcribed data.
> > > > > - **5)** Indicates that pretraining with inpainting without alignment learning is not effective for achieving low WER under low-resource conditions, even with close-to-perfect transcriptions.
> > > > > - **6)** Shows that our proposed pretraining strategy achieves both low WER and high SECS-O, using **1/5** of the pretraining data and **1/100** of the transcribed data compared to **2)**.
> > > > > - **6)** Highlights that our pretraining strategy converges significantly faster compared to inpainting (see **2)** and **5)**).
> > > > > - **7)** Compared to **3)** and **4)**), Using a similar or even smaller amount of transcribed data, a noticeable difference in SECS-O is observed.
> > > > >
> > > > > ---
> > > > >
> > > > > **+)** With this pretraining, the ability to perform effective zero-shot VC compared to existing zero-shot VC methods like SelfVC, which already deliver good performance, is an advantage that can be achieved without any fine-tuning.
> > > > >
> > > > > ---
> > > > >
> > > > > Considering the WER of 2.3 for Ground Truth, we believe the WER falls within the error bar. However, we additionally claimed that A²-Flow is comparable to E2TTS in terms of pronunciation robustness. To support this, as detailed in Table 7 of Appendix A.3, we conducted experiments using only LibriTTS data with 300K iterations of fine-tuning, achieving the same WER of 1.9 and SECS-O of 0.695 as E2TTS.
> > > > >
> > > > > ---
> > > > >
> > > > > The reviewer mentioned that our paper does not provide any generated sample releases. However, we have made not only the full samples used in the subjective evaluation (Table 2) but also additional samples for other languages and voice conversion publicly available on the demo page since the initial submission. The link to the demo page is provided in Appendix A.1. We strongly encourage you to listen to the samples.
> > > > >
> > > > > **Demo Page Link:** [https://anonymous.4open.science/r/demo-page-B24F/index.md](https://anonymous.4open.science/r/demo-page-B24F/index.md)

---

### Official Review · Reviewer_HYvY · 2024-10-31

**Soundness:** 3
**Presentation:** 3
**Contribution:** 3
**Rating:** 8
**Confidence:** 4

**Summary:**

The paper introduces A2-Flow, an alignment-aware pre-training method for flow matching models in speech synthesis. A2-Flow integrates alignment learning directly into the pre-training process using discrete speech units extracted from HuBERT. This approach enables the model to efficiently adapt to alignment-aware tasks such as text-to-speech (TTS) and voice conversion (VC) without the need for separate alignment mechanisms. The authors demonstrate that A2-Flow facilitates alignment-free VC and allows for faster convergence during TTS fine-tuning, even with limited transcribed data. Experimental results show that A2-Flow achieves superior zero-shot VC performance compared to existing models and matches state-of-the-art TTS performance using only a small amount of transcribed data. Additionally, the method is effective in multilingual settings, highlighting its scalability and practical applicability in low-resource scenarios.

**Strengths:**

1. Originality: The paper presents a novel approach by integrating alignment learning directly into the pre-training phase using discrete speech units. This method addresses the limitations of existing models that require external alignment mechanisms or extensive additional training.

2. Quality: Extensive experiments support the effectiveness of A2-Flow. The model demonstrates superior performance in zero-shot voice conversion and matches state-of-the-art results in TTS with significantly less transcribed data.

3. Clarity: The paper is generally well-organized and provides detailed explanations of the methodology, including the integration of discrete speech units and the flow matching framework. The inclusion of diagrams and ablation studies aids in understanding.

4. Significance: A2-Flow offers a practical solution for high-quality speech synthesis in low-resource settings. By reducing the dependency on large amounts of transcribed data, it has substantial implications for multilingual and low-resource language applications.

**Weaknesses:**

1. Limited Baseline Comparisons: While the paper compares A2-Flow with several existing models, a more comprehensive comparison with other state-of-the-art methods, especially in multilingual settings, would strengthen the evaluation.

2. Discussion of Limitations: The paper acknowledges some limitations, such as reliance on self-supervised speech units, but does not deeply explore potential solutions or future work to address these issues.

3. Ablation Study Depth: The ablation studies could be expanded to include more variations in hyperparameters and architectural choices to better understand their impact on performance.

4. Methodology Clarity: Certain aspects of the methodology, like the training details of the total length predictor and the timestep shifting strategy, could be explained in more detail to enhance reproducibility.

**Questions:**

1. Total Length Predictor: Could the authors provide more details on how the total length predictor is trained and integrated into the TTS fine-tuning process?
2. Multilingual Evaluation: Have the authors considered evaluating A2-Flow on additional low-resource languages or dialects to further demonstrate its effectiveness across diverse linguistic settings?
3. Comparison with Alignment-Free Methods: How does A2-Flow compare with other alignment-free speech synthesis methods in terms of computational efficiency and inference speed?

---

> ### Author Response · Authors · 2024-11-28
>
> ### Weakness 1. **Limited Baseline Comparisons & Q2**
>
> - We primarily show the effectiveness of A^2-Flow by leveraging a large amount of English untranscribed data and comparing it with state-of-the-art zero-shot TTS models for English. In multilingual settings, the amount of untranscribed data available for other languages in our experiments was relatively limited. As a result, rather than aiming to propose a state-of-the-art model for other languages, we focused on showing whether a model trained on multilingual untranscribed data could extend alignment learning to other languages with only a small amount of transcribed data. Consequently, we did not conduct direct comparisons with existing state-of-the-art multilingual TTS models. Pre-training our approach on significantly larger and more diverse multilingual untranscribed datasets to achieve state-of-the-art performance in multilingual zero-shot TTS is beyond the scope of this paper and is reserved for future work.
>
> ---
>
> ### Weakness 2. **Discussion of Limitations**
>
> - A%^2%-Flow relies on self-supervised speech units, leveraging HuBERT models in the speech domain. We have explained that our pre-training approach offers advantages over methods like SpeechFlow, which also operate without any conditions, particularly for tasks requiring alignment modeling in the speech domain. However, when extending generative models beyond the speech domain to general audio domains, challenges arise. For instance, the AudioBox paper utilized SpeechFlow, pre-trained without conditions, across both speech and general audio domains. While our A²-Flow approach could potentially be applied in these scenarios, directly using speech-specific self-supervised representations for general audio may not be intuitively appropriate.
>
> ---
>
> ### Weakness 3. **Ablation Study Depth**
>
> - In our experiments, we fixed the architecture design for the reproduced E2TTS and our A²-Flow, and therefore, we did not conduct ablation studies on the architecture itself. However, we provided additional results for A²-Flow-T in Appendix A.3, Table 7, to demonstrate the advantages of the model when using the same amount of transcribed data as E2TTS. Additionally, we included ablation experiments on the impact of different unit representations, showing how the choice of unit representation affects performance in the downstream TTS task.
>
> ---
>
> ### Weakness 4 **Methodology Clarity & Q1**
>
> - We added explanations for the total length predictor and details on timestep shifting in Appendix A.2. We used the ground truth length during fine-tuning for TTS, and the total length predictor was employed only during inference.
>
> ---
>
> ### **Q3**
>
> The DiTTO-TTS and E2TTS models compared in the paper are alignment-free methods. In the DiTTO-TTS paper, the DiTTO-en-L model, which uses the DiT-L architecture similar in size to ours, requires 1.479 seconds (RTF 0.15) to generate 10 seconds of audio. The DiTTO-en-XL model, which uses the DiT-XL architecture and is used for comparison in Table 1 of our paper, requires 1.616 seconds (RTF 0.16) for the same task. The reproduced E2TTS and A²-Flow models in our paper share the exact same architecture, with the only difference being that A²-Flow utilizes unit-based learning. Consequently, the inference speed for both models is identical, achieving an RTF of **0.23** for generating 10 seconds of audio on an A100 GPU, including computations for the total length predictor, flow matching decoder, and vocoder. Compared to VoiceBox and other models with similar architectures, our method generates audio using 32 sampling steps with classifier-free guidance (computing both conditional and unconditional outputs with a batch size of 2) through the first-order Euler’s method. This ensures that the inference time remains reasonably fast.

---

### Official Review · Reviewer_staD · 2024-11-03

**Soundness:** 2
**Presentation:** 4
**Contribution:** 2
**Rating:** 5
**Confidence:** 5

**Summary:**

The paper proposes a new way to pre-train an alignment-aware diffusion-based speech synthesis model. Instead of pre-training the model to inpaint the masked region without any conditions, the authors propose to train the models conditioned on deduplicated discrete HuBERT tokens and force the models to learn the alignment during training. The results show that the models can perform comparably to those without pre-training, all achieved without large-scale transcribed datasets. Additionally, the model can perform voice conversion directly after pe-training with high robustness and similarity to the reference speaker.

**Strengths:**

* **Originality**: The paper proposes a novel way to learn the alignment between phonetic representations and speech and shows that this pre-training scheme can transfer to high-quality text-to-speech synthesis with a small amount of transcribed data, while previous methods that do not incorporate alignment pre-training fails to do.

* **Quality**: The paper has compared various models and conducted extensive experiments to examine the effectiveness of its models against several baselines, including E2TTS, without pre-training in a small amount of transcribed data and multi-lingual settings.

* **Clarity**: The paper has an excellent presentation of its methods and results, avoiding overcompcaliting with mathematical definitions, making it fairly easy to follow.

* **Significance**: The paper has shown that it is possible to achieve comparable performance to the most recent duration-free E2TTS without a large amount of transcribed data.

**Weaknesses:**

The major weakness is the contribution of this work seems incremental and does not demonstrate significant improvement over previous methods based on the argument presented in the paper. It is unclear what this pre-training scheme brings compared to previous methods such as SpeechFlow and E2TTS. There are a few arguments where the proposed alignment-aware pre-training can provide some advantages over previous methods. However, none of these seems to be fully supported by the results in the paper and existing literature:

1. It can be argued that this pre-training scheme allows simple transfer learning without transcription of speech data for high-quality TTS. However, most existing large-scale TTS models do not rely on human-labeled data but instead use automatically transcribed data. For example, Chen et al. 2024 [1] trained E2TTS on Emilia [2], which is a 100k-hour multilingual dataset with transcriptions obtained through the ASR model WhisperX. Chen et al. 2024 show that E2TTS trained on ASR-transcribed data can obtain a WER of 2.95, which is not significantly different from 2.2 in the paper. Similarly, NaturalSpeech 3 [3] has achieved a WER of 1.81, even lower than ground truth with ASR-transcribed data. It is unclear whether this method is better than training directly on ASR-transcribed data from a practical point of view, either in terms of quality or training resources. For example, in Figure (b), E2TTS trained on transcribed data only needs to be trained for 700k steps to have comparable performance of A2-Flow pre-trained for 700k steps plus 150k fine-tuning steps. Since E2TTS can perform well on the ASR-transcribed dataset, this suggests that A2-Flow takes more time and resources for training while achieving similar performance as E2TTS. It would be helpful for the authors to demonstrate other performance advantages of A2-flow, such as comparing it to E2TTS trained on an ASR-transcribed dataset (which can also be considered unlabeled), or comparing the total amount of time and resources required for training (i.e., to show that the entire pipeline of pre-training + fine-tuning of A2-flow is more efficient than transcribing + training for E2TTS).

2. It can be argued that this pre-training scheme allows for transferring learning in low-resource settings where transcribed data is limited. However, this advantage is not well supported in the paper either. It is well known that fine-tuning for a different language requires much less data than training it from scratch [4,5]. It is unclear whether this method is better than training E2TTS on a transcribed English dataset (which is enormous) and then fine-tuning it on a low-resource dataset (for example, French with 280 hours of data).

3. It can be argued that this method removes alignment labels for TTS models compared to SpeechFlow. However, it is unclear whether the alignment labels are hard to learn to begin with. Forced alignment can be obtained with a CTC-based ASR model, and it is shown in HuBERT and WavLM that CTC-based ASR models can be trained with sufficiently high quality with limited labeled data (960 hours). The contribution of this work can be further enhanced if the author shows that SpeechFlow fine-tuned on CTC-based alignments obtained from a fine-tuning HuBERT on 960 hours of data is worse compared to A2-Flow without pre-defined alignments.

In addition, the subjective evaluations are rather weak. The authors only compared 19 samples of E2TTS and 9 samples of SpeechFlow and concluded that the model is comparable to E2TTS and better than SpeechFlow in Table 2, and no statistical significance is reported for this result. Most recent papers [1, 3] use at least 40 samples (including the E2TTS paper), so 9 samples are insufficient to establish statistical significance. Moreover, no subjective evaluation is conducted for naturalness but it is a common practice for evaluating the true performance of speech synthesis models, as most current objective evaluation metrics only correlate weakly with MOS. Since the authors claim to have reproduced E2TTS and SpeechFlow in the paper, the paper's argument can be further enhanced by comparing to more samples of E2TTS and SpeechFlow, even with reproduced models.

**References**

[1] Chen, Y., Niu, Z., Ma, Z., Deng, K., Wang, C., Zhao, J., ... & Chen, X. (2024). F5-TTS: A Fairytaler that Fakes Fluent and Faithful Speech with Flow Matching. arXiv preprint arXiv:2410.06885.

> Note: This work was published after the ICLR deadline. However, since this work contains reproduced results from a cited paper (E2TTS), I believe this reference is relevant to this review.

[2] He, H., Shang, Z., Wang, C., Li, X., Gu, Y., Hua, H., ... & Wu, Z. (2024). Emilia: An extensive, multilingual, and diverse speech dataset for large-scale speech generation. arXiv preprint arXiv:2407.05361.

[3] Ju, Z., Wang, Y., Shen, K., Tan, X., Xin, D., Yang, D., ... & Zhao, S. (2024). Naturalspeech 3: Zero-shot speech synthesis with factorized codec and diffusion models. arXiv preprint arXiv:2403.03100.

[4] Lux, F., Koch, J., & Vu, N. T. (2022). Low-resource multilingual and zero-shot multispeaker TTS. arXiv preprint arXiv:2210.12223.

[5] Debnath, A., Patil, S. S., Nadiger, G., & Ganesan, R. A. (2020, December). Low-resource end-to-end sanskrit tts using tacotron2, waveglow and transfer learning. In 2020 IEEE 17th India Council International Conference (INDICON) (pp. 1-5). IEEE.

**Questions:**

1. Is the HuBERT used in the paper trained in English only? If so, how does it work to reconstruct speech in other languages? Which layer is used?

2. What is the model used to compute SECS-O? Is it the WavLM-ECAPA large fine-tuned?

3. Why only conduct subjective evaluations on 19 and 9 samples even though you have reproduced E2TTS and FlowSpeech? Why is there no subjective evaluation of naturalness?

**Details Of Ethics Concerns:**

There is no information about the crowdsourcing platform, demographics, compensation, or criteria for hiring human subjects for the subjective evaluation. Please add this information to the appendix. Also, please indicate whether you have obtained any IRB approval for these evaluations.

---

> ### Author Response · Authors · 2024-11-27
> **Response 1/2**
>
> ### Weaknesses 1.
> - **Regarding ASR Transcribed Data for High-Quality Zero-Shot TTS**
>
> While it is possible to construct transcribed datasets for TTS using ASR systems, this approach assumes the ASR model is highly accurate and produces reliable transcriptions. ASR models often struggle with segmentation issues, such as mid-word cuts, which significantly impact performance. Additionally, constructing datasets like the Emilia dataset in F5-TTS requires complex preprocessing steps, such as ASR using Whisper and WhisperX, followed by filtering based on language ID, speech quality, and average character duration.
>
> While effective for English, this process becomes more challenging for languages with less accurate ASR models. Additionally, even for English, Whisper outputs may include hallucinations. According to the mentioned F5-TTS paper, training E2TTS on the Emilia dataset resulted in a WER of 2.95, compared to 1.9 (with $\alpha=3$) and 2.1 (with $\alpha=1$) for our reproduced E2TTS, likely due to transcription inaccuracies. Our proposed A²-Flow offers an alternative by using HuBERT units for pre-training, eliminating the need for ASR or transcriptions. Notably, both the HuBERT units and pre-training data are entirely untranscribed.
>
> This approach enables competitive TTS performance with minimal transcribed data, as demonstrated for English. We believe it offers significant benefits for communities with large-scale untranscribed data but limited transcription resources, enabling high-quality zero-shot TTS for under-resourced languages.
>
> - **Regarding Training Resources:**
>
> A²-Flow and E2TTS were trained using 32 GPUs for pre-training, while A²-Flow fine-tuning on LibriTTS utilized only 8 GPUs. Calculating GPU hours, 150K fine-tuning iterations on 8 GPUs equates to 37.5K iterations on 32 GPUs. For transparency, we share the results for A²-Flow fine-tuned for 150K iterations (8 GPUs) and E2TTS trained for 700K and 850K iterations (32 GPUs). The experimental results are detailed in Appendix A.3, Table 7.
>
> |Model|Training Details|WER|SECS-O|UTMOS|
> |---|------|-----|-----|----|
> |A²-Flow | 700K PT (32 GPUs, 40K hours) + 150K FT (8 GPUs, 500 hours) | 2.2| 0.704| 4.03|
> |A²-Flow-T| 700K PT (32 GPUs, 40K hours) + 150K FT (32 GPUs, 40K hours) | 2.0| 0.711| 4.01|
> |E2TTS| 700K Train (32 GPUs, 40K hours) |1.9|0.694| 4.01|
> |E2TTS| 850K Train (32 GPUs, 40K hours) |2.0|0.695| 4.02|
>
> As shown, training E2TTS for an additional 150K iterations (total 850K) resulted in negligible improvement compared to 700K iterations. This demonstrates that A²-Flow’s results were not achieved simply by training longer. Additionally, A²-Flow fine-tuning required fewer GPUs and less transcribed data than E2TTS while achieving superior SECS-O performance.
>
> Furthermore, we provide results for A²-Flow-T, which refers to A²-Flow fine-tuned on 40K hours of transcribed data (the same amount used for E2TTS) using 32 GPUs. With the same computational resources as E2TTS at 850K iterations, A²-Flow achieved a WER of 2.0 and a SECS-O of 0.711, surpassing E2TTS in SECS-O performance. These results demonstrate that even with fully transcribed datasets, the alignment-aware pre-training of A²-Flow enhances zero-shot TTS learning, creating a synergistic effect that significantly boosts performance.
>
> Additionally, our pre-training method not only demonstrates boosted performance but also shows that the pre-trained model itself enables high-quality zero-shot VC without the need for fine-tuning.
>
> ----
>
> ### Weaknesses 2.
> It is true that a pre-trained English model can be fine-tuned with limited transcribed data for other languages and achieve reasonable results. As shown in the table above (and in Table 7 of the paper), A²-Flow demonstrates superior performance compared to E2TTS under equivalent conditions, particularly in SECS-O metrics, when fine-tuned with the same amount of transcribed data. This underscores the advantage of A²-Flow in scenarios with limited transcribed data.
>
> However, A²-Flow offers a unique benefit beyond the ability to fine-tune from transcribed data: it leverages untranscribed data effectively to enhance zero-shot TTS performance. For languages where transcribed data is scarce but large amounts of untranscribed data are available, A²-Flow provides a robust method to utilize such resources. In contrast, traditional approaches like E2TTS heavily depend on transcriptions and face challenges in incorporating untranscribed data to boost performance.
>
> By employing untranscribed data during the pre-training stage, A²-Flow offers a scalable and practical solution for low-resource languages. This capability significantly enhances zero-shot TTS performance without requiring extensive transcription resources, addressing a critical gap in zero-shot TTS development and broadening its applicability to under-resourced language communities.

---

> > ### Author Response · Authors · 2024-11-27
> > **Response 2/2**
> >
> > ### Weakness 3.
> >
> > We are not claiming that obtaining alignment labels is inherently difficult. However, when training SpeechFlow using alignment labels as aligned phonemes, as described in the SpeechFlow paper, it requires a duration predictor to align each phoneme with the corresponding speech. Methods that rely on a duration predictor and explicitly input alignment phonemes, such as VoiceBox, have been shown in the E2TTS paper to underperform compared to approaches like E2TTS, which internally learn alignments without the need for predefined labels. Therefore, in our work, we focused on comparing our method with a stronger baseline, E2TTS, which represents a more advanced and effective framework for alignment learning. We focused on demonstrating through Figure 2 that the SpeechFlow approach is incompatible with the E2TTS framework, which learns alignments intrinsically.
> >
> > ---
> >
> > ### Regarding Subjective evaluation & Q3.
> >
> > Since most zero-shot TTS research models are not open source, comparisons are primarily conducted using objective metrics like ASR-based WER and speaker similarity metrics (SECS-O), measured over a large number of samples in the test set. When models are not publicly available, subjective evaluations are limited to the samples provided on the demo pages of the baseline models, which is a common practice. While subjective metrics measured on a small number of samples cannot provide statistically significant results compared to objective metrics based on 1000+ samples, they are offered as a proxy. For example,  in the F5-TTS [1] paper, no subjective evaluation results are provided for the LibriSpeech test-clean set, and in the NaturalSpeech 3 [3] paper, CMOS was evaluated using 20 randomly selected samples and SMOS using 10 samples.
> >
> > In our paper, we conducted subjective evaluations using all 9 samples available from the SpeechFlow demo page and 19 samples provided on the E2TTS demo page, which include LibriSpeech test-clean samples and hard sentence samples. Additionally, for these evaluations, we had 100 native English speakers rate every sample pair, resulting in a total of 900 evaluations for SpeechFlow and 1,900 evaluations for E2TTS. While the number of comparable samples is limited, the total number of evaluations is substantial due to the comprehensive comparisons conducted by all 100 evaluators.
> >
> > We acknowledge the lack of naturalness evaluation in the original submission and appreciate the reviewer’s suggestion. To address this, we have included UTMOS, an automatic metric that approximates MOS, as a measure of naturalness. The average UTMOS score for the entire test set was added to our results. Compared to the ground truth UTMOS average of 4.10, our model achieved a score above 4.0, similar to the UTMOS score of the reproduced E2TTS, as shown in Table 6 and 7.
> >
> > ---
> >
> > ### Q1.
> >
> > We utilized the open-source version of the HuBERT model available at https://github.com/facebookresearch/textlesslib/tree/main/examples/expresso. As described in the paper, this HuBERT model was trained on Common Voice, VoxPopuli, and Multilingual LibriSpeech datasets. The 12th layer was used.
> >
> > We've also included results in Table 8 using different unit representations (HuBERT-base trained on LibriSpeech, K=200). These results demonstrate that the proposed method performs well even with varying HuBERT unit representations.
> >
> > ---
> >
> > ### Q2.
> >
> > We used the WavLM-ECAPA large model, the same as the one used in VALL-E, to ensure consistency with the baseline evaluations.
> >
> > ---
> >
> > Regarding human evaluation, we've provided detailed information about human evaluation in Appendix A.5.

---

> > ### Comment · Reviewer_staD · 2024-11-30
> >
> > I appreciate the authors' articulated rebuttal. However, I believe the responses do not address my concerns fully. Here are my responses to the rebuttal:
> >
> > **Regarding ASR Transcribed Data for High-Quality Zero-Shot TTS**
> >
> > NaturalSpeech 3 used transcribed data from LibriLight with ASR and achieved a WER of 1.81%, which is lower than A$^{2}$-Flow and E2-TTS altogether. It is possible that this approach will introduce some improvement in intelligibility. However, since there is no experiment conducted specifically to test whether ASR-transcribed data results in higher WER, we can't conclude directly that A$^{2}$-Flow is better than ASR-transcribed data, as NaturalSpeech 3 obtained a WER of 1.81% with transcribed data.
> >
> > **Regarding Training Resources**
> >
> > These do not really address my concern, which is mainly about extra training resources for fine-tuning. My concern is that A$^{2}$-Flow requires extra fine-tuning steps that may consume more resources, and the performance is not necessarily better (similar UT-MOS, slightly higher WER, and slightly higher SIM). All of these evaluations are objective and hard to test for statistical significance. We need subjective evaluations from human raters to establish statistically significant better performance.
> >
> > **Regarding Weakness 2**
> >
> > This approach only makes sense for limited resources. The table above does not show the ability to transfer to different languages (compared to E2-TTS). There is no experiment that compares the performance between A$^{2}$-Flow and E2-TTS of new languages by fine-tuning on limited transcribed data of low-resource languages. Since ICLR 2025 now prohibits new experiments after Nov 26th, I don't think there is a way to address this weakness now.
> >
> > **Regarding Subjective Evaluation**
> >
> > I don't think this justifies the use of limited samples for MOS evaluations. UT-MOS does not correlate with MOS perfectly, and it is not possible to conclude statistical significance with only 9 samples, no matter how many raters were employed, as these samples could be biased and limited in testing cases and not trained on the same training data either. Since you have already reproduced the models with the same training data, why not just test more samples using reproduced models for both similarity and naturalness?

---

> > > ### Author Response · Authors · 2024-12-04
> > >
> > > **Regarding ASR Transcribed Data for High-Quality Zero-Shot TTS**
> > >
> > > Regarding NaturalSpeech 3, while direct numerical comparison is difficult since it evaluates one randomly sampled utterance per speaker in the test set instead of the entire sample set, the E2TTS paper, authored by the same team as NaturalSpeech 3, reports a WER of 2.6 and SECS-O of 0.632.
> > >
> > > Our methodology is more appropriate to compare with the WER results (2.95) of E2TTS, which is directly evaluated in the F5-TTS paper suggested by the Reviewer as related work, rather than with NaturalSpeech 3, which uses a duration predictor. Specifically, F5-TTS, trained with a larger amount of English data using Whisper, reproduces the pronunciation accuracy of E2TTS. In contrast, our approach achieves comparable pronunciation accuracy to the Ground Truth WER of 2.3 using only 500 hours of transcribed data—equivalent to 1% of the total—after pre-training, without requiring any filtering or transcription process for unlabeled data.
> > >
> > > And what we want to emphasize is that creating a refined transcription dataset for unlabeled data is not a straightforward process and requires significant effort, as evidenced by datasets like LibriHeavy or Emilia, which were detailed in research papers. Since constructing such transcribed data is far from trivial, we hope it is acknowledged that our method achieves similar performance without the need to account for ASR transcription errors or implement filtering methods.
> > >
> > > **Regarding training resources**
> > >
> > > Our proposed method employs a pre-training approach using only unlabeled data, which inevitably requires an additional fine-tuning process. However, the fine-tuning process (150K steps, 8 GPUs) accounts for approximately 5.3% of the computational effort compared to training E2TTS from scratch (700K steps, 32 GPUs) or pre-training A²-Flow (700K steps, 32 GPUs), based on GPU-hours (150/700 * 4 = 5.3%). The fine-tuning process can be completed in 1.2 days using 8 GPUs, meaning it does not require a significant increase in training resources.
> > >
> > > Additionally, our pre-training method enables voice conversion functionality without any fine-tuning, and it achieves better SIM and comparable WER using only 1% of transcribed data, which is a significant advantage. As highlighted in our previous response, with the same amount of transcribed data, our approach shows a noteworthy difference in SECS-O while utilizing a similar amount of computational resources as E2TTS.
> > >
> > > **Regarding Weakness 2**
> > >
> > > A$^2$-Flow pre-training demonstrates that by leveraging a large amount of unlabeled data in English for pre-training, it is possible to achieve strong performance using only limited amount of labeled data. However, this simulates a scenario where a large amount of unlabeled data is available for English, and it can be applied universally to any other language without losing generality, regardless of whether ASR performs well for that language or not. While it is also feasible to fine-tune using a limited amount of transcribed data with an English model, this method does not fully utilize the untranscribed data available in the target language. Thus, we highlight that these two approaches are compatible and complementary.
> > >
> > > **Regarding Subjective Evaluation**
> > >
> > > We have provided the results for the subjective evaluation in the official comment.

---

### Official Review · Reviewer_98gk · 2024-11-04

**Soundness:** 2
**Presentation:** 3
**Contribution:** 2
**Rating:** 6
**Confidence:** 5

**Summary:**

A2-Flow is a zero-shot TTS model based on flow matching, which relies on self-supervised speech units from HuBERT for pre-training to implicitly align speech and duration. Subsequently, A2-Flow can perform speech synthesis without the need for separate alignment mechanisms  through fine-tuning with a small amount of text-speech paired data. Due to pre-training, A2-Flow can achieve alignment-free voice conversion and allows for faster convergence during TTS fine-tuning.

**Strengths:**

A2 Flow proposes using semantic tokens derived from unlabeled data for pre-training, aiming to align speech and text. This approach allows for zero-shot text-to-speech (TTS) synthesis using only a small amount of paired text-speech data, eliminating the need for separate alignment mechanisms. Additionally, due to the presence of pre-trained alignment, alignment-free voice conversion becomes possible.

**Weaknesses:**

The paper is somewhat helpful for community advancement, but I believe additional experiments could further demonstrate its validity and effectiveness. For example, some experiments like MOS, CMOS, and SMOS could be added to prove its effectiveness from a subjective perspective.

**Questions:**

- Can you explain how your overall speech duration predictor is designed, and whether it shows good robustness for shorter and longer texts? Additionally, when dealing with longer or shorter texts, is the alignment between your speech and text still stable?
- As iterations increase, will your WER and SECs-O metrics maintain a good level? Will there be a tendency to forget the alignment mechanism, or will they align with the E2TTS level? In this setting, will the improvements from pretraining still be superior to models like E2TTS?
- I've noticed that after feature extraction with HuBERT, there is a de-duplication step. Is this step intended to further align the text and semantic tokens? Would removing this step have a significant impact on the results?

---

> ### Author Response · Authors · 2024-11-27
>
> ### Weaknesses 1. **Additional Experiments**
>    - In the paper, we provided objective metrics (WER, SECS-O) for each model but did not include a separate evaluation of audio quality. However, in Table 2, we presented the results of A/B tests comparing our model with SpeechFlow and E2TTS based on human evaluations, focusing on similarity to reference audio in terms of "prosody, emotion, and timbre," instead of providing CMOS or SMOS. As the baseline models are not open-source, we compared our model to samples available on their demo pages, consistent with prior works.
> - Additionally, we have included results evaluating speech naturalness to supplement the existing subjective evaluation and objective metrics. To assess speech naturalness, we provide UTMOS results for all samples in the LibriSpeech test-clean evaluation. UTMOS, which is known to correlate highly with human subjective ratings, serves as a robust proxy for audio naturalness. The average UTMOS scores are included in Tables 6, 7, and 8 in Appendix A.3.
> - Furthermore, as requested, we have added a comparison of model performance over different training iterations in Table 7, directly comparing the results with E2TTS. Additionally, to demonstrate the effectiveness of the pre-training method with different unit representations, we have included experimental results in Table 8, highlighting the impact of various unit representations.
>
> ---
> ### Q1-1) **Speech Duration Predictor Design and Robustness**
> We detailed the total length predictor's design in Appendix A.2.1 and confirmed its robustness for shorter and longer texts. Using samples from LibriSpeech test-clean, we computed the mean and 95% confidence intervals of the ratio between predicted length and ground truth length for various durations:
>
> | Duration (seconds) | Mean Ratio | 95% Confidence Interval |
> |----------|------------|-------------|
> | 3–4 seconds |1.119| ± 0.015|
> | 4–10 seconds|1.018| ± 0.009|
> | 10–15 seconds|1.021 | ± 0.014|
>
> The results demonstrate that the total length predictor performs robustly across varying text lengths, outputting ratios close to 1. For very short text inputs, we concatenate the prompt text with the target text sequence to predict the duration. Sample results for these cases are provided in the **demo page**.
>
> ### Q1-2) **WER Performance by Text Length**
> The WER and its 95% confidence interval for each duration range were computed as follows:
>
> | Duration (seconds) | WER (%) | 95% Confidence Interval |
> |----------------|---------|--------|
> | 3–4 seconds |2.99| ± 0.010 |
> | 4–10 seconds |2.23| ± 0.003|
> | 10–15 seconds|3.51| ± 0.006|
>
> Although WER slightly worsens for shorter or longer durations compared to the 4–10 second range, the alignment remains stable, maintaining good WER performance.
>
> ---
> ### Q2) **Impact of More Iterations for Fine-tuning**
> We extended the fine-tuning process to 300K iterations, adjusting the learning rate decay schedule to reach zero at the end. The results are provided in **Table 7 (Appendix A.3)**. Below is a summary:
>
> | Model                               | WER   | SECS-O | UTMOS |
> |--------------------|-------|-------|-------|
> | A$^2$-Flow (700K pre-train, 150K FT)| 2.2| 0.704 | 4.03  |
> | A$^2$-Flow (700K pre-train, 300K FT)| 1.9| 0.695 | 4.06  |
> | E2TTS (700K)|1.9| 0.696 | 4.01|
> | E2TTS (850K)|2.0| 0.695 | 4.02|
>
> Increasing fine-tuning to 300K iterations improves text-speech alignment, reducing WER to 1.9, comparable to E2TTS. However, SECS-O slightly declines from **0.704** (150K FT) to **0.695** (300K FT), likely due to overfitting to LibriTTS. Despite this, A$^2$-Flow’s SECS-O remains comparable to E2TTS, which uses significantly more transcribed data for training. These results demonstrate that A$^2$-Flow effectively maintains strong alignment and TTS performance with extended fine-tuning, even outperforming E2TTS in WER with fewer resources.
>
> ---
> ### Q3) **Impact of the De-duplication Step in HuBERT Processing**
> The de-duplication step in our method is essential for learning the alignment between input sequences and speech. While HuBERT unit sequences are inherently aligned with speech, removing consecutive repetitions creates de-duplicated units that focus on semantic transitions. This enables the model to learn both alignment and duration information internally during pre-training.
>
> Without de-duplication, the model would process repeated tokens, making it difficult to learn alignment or adapt effectively to downstream tasks like TTS, where aligning text and speech is critical. The absence of this step would also hinder fine-tuning in scenarios without explicit duration information, as the model would lack the foundational alignment learned during pre-training.
>
> Thus, de-duplication ensures that the pre-training closely mimics alignment challenges in TTS, making it a core component of our alignment-aware pre-training approach.

---

### Author Response · Authors · 2024-11-27
**Official Comment**

We sincerely thank all reviewers for their constructive feedback and positive evaluation of our work. Based on your suggestions, we have conducted additional experiments and provided detailed clarifications to strengthen our paper. Below, we summarize our updates and responses to the reviewers' comments:

Additional Experimental Results:

* We have included the results of additional experiments requested by the reviewers in Tables 6, 7, and 8 (Appendix). Specifically:
To evaluate speech naturalness, we used UTMOS, a model trained to predict MOS as a proxy for naturalness, and reported the average UTMOS score for the entire evaluation set (over 1000 test samples). This metric is now included alongside existing evaluation metrics in Tables 6, 7, and 8, providing a robust and reproducible measure of speech naturalness.
* Table 7 addresses Reviewer staD’s concern that our results might stem from additional fine-tuning iterations compared to E2TTS. To clarify, we provide a comparison of E2TTS and A$^2$-Flow across various training iterations, including a configuration where A$^2$-Flow is fine-tuned with the same amount of transcribed data (40K hours) as E2TTS, referred to as A$^2$-Flow-T. The results demonstrate that A$^2$-Flow-T achieves superior SECS-O scores compared to both the original A$^2$-Flow-Flow and E2TTS.
* Table 8 presents an ablation study to assess the impact of different discrete units used during pre-training. The results show that our pre-training method remains effective when fine-tuned on LibriTTS, regardless of the unit representation, highlighting the robustness of our approach.

Detailed Methodological Clarifications:

* Total Length Predictor: In response to comments from Reviewer 98gk and Reviewer HYvY
, we have added a detailed explanation of the Total Length Predictor’s structure and specifications in Section A.2.1 of the Appendix.
* Timestep Shifting: As suggested by Reviewer HYvY
, we have expanded our explanation of timestep shifting in Section A.2.2, supported by Figure 4 and Algorithm 1 for clarity.
* Human Evaluation: At the suggestion of Reviewer staD, we have provided additional details about our human evaluation methodology, including compensation, evaluator selection, and evaluation criteria, in Appendix A.5.

Core Contributions: To reiterate the core contributions of our work:

* Background: E2TTS, which learns text-speech alignment without a phoneme duration predictor, outperforms most zero-shot TTS models. However, E2TTS requires significant computational resources, large amounts of transcribed data, and extensive training iterations, making them challenging in resource-constrained settings.
* Proposed Approach: We propose an alignment-aware pre-training method that effectively initializes TTS models within the E2TTS framework to model internal text-speech alignment without requiring a duration predictor.

Our contributions are as follows:
* We demonstrate that using de-duplicated unit sequences instead of text during pre-training enables efficient alignment learning and adaptation to limited paired <text, speech> data during fine-tuning.
* We show that our pre-training method enables high-quality zero-shot voice conversion without requiring additional fine-tuning.
We highlight the language-agnostic nature of our method, demonstrating its applicability across multiple languages.
* Compared to state-of-the-art zero-shot TTS models, our approach achieves superior zero-shot TTS performance using just 1% of the transcribed data. Additionally, it is the first pre-training approach to achieve such results while learning internal alignment without relying on a phoneme duration predictor.

We appreciate the reviewers' valuable feedback, which has helped us improve the clarity and rigor of our work. Thank you for your time and effort in reviewing our submission.

---

### Author Response · Authors · 2024-12-04
**Providing subjective evaluation results (CMOS, SMOS)**

The discussion period is nearly over, but we would like to provide the subjective evaluation requested during the rebuttal process and hope that reviewers can refer to it in their final evaluation.

In our initial submission, we compared our model with all 9 samples from the SpeechFlow demo page and 19 samples from the E2TTS demo page for direct comparison with the benchmark models. In situations where subjective evaluation with models that are not open-sourced is limited, we followed the same approach as existing published papers, comparing our results with the samples available on the demo pages to ensure a direct comparison with the official models rather than reproduced ones. (We would like to reiterate that our subjective evaluation compared to the state-of-the-art E2TTS paper is not significantly lacking, considering that NaturalSpeech 3 uses 20 samples for CMOS and 10 samples for SMOS.)  Furthermore, as a proxy for the subjective evaluation of speech naturalness, we provided UTMOS results for all samples in the test set, comprising over 1,000 samples, during the rebuttal process.

Furthermore, to strengthen the subjective evaluation results of the paper, we have added additional subjective evaluation in the official comment. These subjective evaluations compare our model with reproduced baselines following **Reviewer staD**’s requests. The baselines are as follows:

1. **E2TTS (SpeechFlow pre-training)**: SpeechFlow pre-trained with 40K hours of unlabeled data and fine-tuned on 500 hours of LibriTTS data using the E2TTS method. This corresponds to "SpeechFlow-E2" in Table 1 of the original paper.
2. **E2TTS (Baseline)**: The reproduced E2TTS model trained with 40K hours of data over 700K steps.

For CMOS and SMOS subjective evaluations, we used Amazon Mechanical Turk as an additional platform. 50 random samples were selected from the LibriSpeech test-clean dataset used for the paper's evaluation, and these were used for comparative evaluation.

For the CMOS evaluation, 100 raters were asked to evaluate three random samples out of the 50, comparing two models on a scale of -3 to 3. The instruction given was: "**How do the two audio samples compare in terms of pronunciation accuracy and speech quality?**" Transcripts were provided, and the samples from the two models were presented in a random order for each evaluation.

For the SMOS evaluation, 150 raters were asked to evaluate three random samples. A reference audio was provided along with the generated sample, and raters scored the similarity on a scale of 1 to 5 based on the instruction: "**How similar are the given audio samples to the provided reference speech in terms of prosody and the speaker's voice characteristics?**" Transcripts were not provided during SMOS evaluations.

In the CMOS evaluation, positive scores indicate that E2TTS (A²-Flow pre-training) outperforms the baseline, while negative scores indicate inferior performance. For SMOS, higher scores indicate greater speaker similarity.

### Results: Subjective and Objective metrics on LibriSpeech test-clean dataset

| Model                            | Unlabeled | Labeled | **CMOS**               | **SMOS**  | WER | SECS-O|
|----------------------------------|--|--|-----|--------------------|--------------------|-----|
| E2TTS (A²-Flow pre-training)     |40000|500  | **0.0**                | **3.84 ± 0.08**        | 2.2 | 0.704 |
| E2TTS (SpeechFlow pre-training)  |40000|500     | **-1.31 ± 0.19**      |**3.70 ± 0.08**       | 107.7 | 0.690
| E2TTS                            |0|40000     |**-0.19 ± 0.16**    | **3.68 ± 0.08**        | 1.9 | 0.696|

According to the results, as shown in Table 2, the **A²-Flow method demonstrated higher performance in SMOS, similar to the observed difference in SECS-O compared to the SpeechFlow pre-training method and E2TTS trained from scratch.** While E2TTS (SpeechFlow pre-training) achieved a high SECS-O score due to its strong speaker similarity to the reference in objective metrics, its SMOS score was comparable to that of E2TTS. However, it failed to learn alignment, resulting in a WER exceeding 100. Similarly, **its CMOS score showed a significant deficiency compared to the A²-Flow pre-training method.** Compared to E2TTS, the **A²-Flow pre-training method achieved a slightly higher CMOS score**, further reinforcing the validity of A²-Flow as a meaningful pre-training approach based on this subjective evaluation.

We would be truly grateful if these results and the responses provided to each reviewer are taken into account during the final evaluation.

---

### Meta-Review · Area_Chair_SkP1 · 2024-12-19

**Metareview:**

A2 Flow introduces an approach by using semantic tokens derived from unlabeled data for pre-training, aligning speech and text. This enables zero-shot text-to-speech (TTS) synthesis with minimal paired text-speech data and allows for alignment-free voice conversion. The integration of alignment learning directly into the pre-training phase using discrete speech units addresses the limitations of existing models that require external alignment mechanisms or extensive additional training. The paper is well-organized, providing detailed explanations of the methodology, including the integration of discrete speech units and the flow matching framework. Diagrams and ablation studies enhance understanding.

However, the work appears incremental and does not show significant improvement over previous methods like SpeechFlow and E2TTS. A more comprehensive comparison with other state-of-the-art methods, especially in multilingual settings, would strengthen the evaluation. The paper acknowledges some limitations, such as reliance on self-supervised speech units, but does not deeply explore potential solutions or future work to address these issues. The ablation studies could be expanded to include more variations in hyperparameters and architectural choices to better understand their impact on performance.

**Additional Comments On Reviewer Discussion:**

Despite the authors addressing the reviewers' questions and raising awareness of the paper's contributions, only one reviewer increased their score. The other two reviewers maintained their original scores, as their concerns were not fully resolved. Consequently, the average overall rating remains below the acceptance threshold.

---

### Decision · Program_Chairs · 2025-01-22

Reject